# Modeling Training Dynamics and Error Estimates of DNN-based PDE Solvers: A Continuous-time Framework

## Abstract

Deep neural network-based PDE solvers have shown remarkable promise for tackling high-dimensional partial differential equations, yet their training dynamics and error behavior are not well understood. This paper develops a unified continuous-time framework based on stochastic differential equations to analyze the noisy regularized stochastic gradient descent algorithm when applied to deep PDE solvers. Our approach establishes weak error between this algorithm and its continuous approximation, and provides new asymptotic error characterizations via invariant measures. Importantly, we overcome the restrictive global Lipschitz continuity loss gradient, making our theory more applicable to practical deep networks. Specifically, our study focuses on general second-order elliptic PDEs; however, the proposed framework is not limited to this specific form and can be extended in principle to broader classes of PDEs. Furthermore, we conduct systematic experiments to reveal how stochasticity affects solution accuracy and the stability domains of optimizers. Our results indicate that stochasticity can have varying impacts on the stability of solutions near different local minima; therefore, in practical training, strategies should be dynamically adjusted according to the local optimization landscape to enhance robustness and stability of neural PDE solvers.

## 1 Introduction

Partial differential equations are essential tools for modeling phenomena across physics, biology, and engineering, yet classical numerical methods like finite element and finite difference schemes often struggle with the curse of dimensionality in high-dimensional settings. Recently, deep neural networks (DNNs) have emerged as powerful alternatives for approximating PDE solutions, with physics-informed neural networks (Sirignano & Spiliopoulos, 2018; Raissi et al., 2019) drawing particular interest by embedding the governing equations directly into the loss function. Other notable approaches include the Deep Ritz method (Yu & E, 2018), which leverages the variational formulation of PDEs, and the Weak Adversarial Networks framework (Zang et al., 2020), which utilizes the weak form to effectively address complex boundary conditions and irregular domains.

However, theoretical understandings of training dynamics for these PDE solvers remains elusive. This challenge arises because the training process, especially under stochastic optimization algorithms, is profoundly influenced by noise and the high dimensionality of the parameter space, both of which significantly affect convergence and stability (Ge et al., 2015; Keskar et al., 2022). To inform our understanding of these dynamics, we take inspiration from theoretical developments in supervised learning, where the analysis of training behavior is more advanced. In recent years, a popular approach in supervised learning has been to employ continuous-time models to study optimization dynamics (Dai & Zhu, 2020; Li et al., 2017). These models, however, often depend on stringent assumptions such as global Lipschitz continuity of the loss function, which are typically valid only for linear networks, such as random feature models. Furthermore, loss functions in PDE solvers are considerably more complex than those in supervised learning, further limiting the applicability of existing continuum theories. As a result, prior continuous-time models offer, at best, heuristic guidance, underscoring the need for new theoretical frameworks tailored to the unique complexities of neural network-based PDE solvers.

In this paper, we develop a continuous-time framework to model the local training dynamics of deep neural network–based PDE solvers, using PINNs as a representative case. This framework dispenses with global Lipschitz assumptions and offers a new perspective on error estimation via continuous-time modeling. We also conduct systematic experiments to examine how stochasticity in optimization algorithms influences training dynamics and performance.

## 1.1 RELATED WORKS

**Continuous-time modeling of stochastic optimization methods.** Continuous-time formulations of stochastic gradient descent have emerged as a powerful tool for analyzing the training dynamics of neural networks. By approximating SGD by stochastic differential equations, researchers have gained deeper insights into optimization trajectories and convergence behavior. Notably, Chaudhari & Soatto (2018) and Li et al. (2017) established SDE-based analyses with weak convergence results for supervised learning. Building on this, Hu et al. (2019) explored diffusion approximations for non-convex SGD, revealing the essential role of stochasticity in escaping unstable stationary points. Further, Dai & Zhu (2020) employed the Fokker-Planck equation to show how batch size can influence the sharpness of minima, while Smith & Le (2018) linked batch size to generalization from a Bayesian perspective. Continuous-time models have also been developed for dropout algorithm (Zhang et al., 2024). However, most existing studies are limited to supervised learning, where loss landscapes are typically less complex and more structured than those in PDE solvers.

**Training theory and error estimation in DNN-based PDE solvers.** Understanding the training dynamics and stability of DNN-based PDE solvers remains challenging, largely due to the stochasticity of gradient-based optimization in high-dimensional parameter spaces. Early works such as Mei et al. (2018) and Chizat et al. (2019) analyzed mean-field dynamics and highlighted the role of noise in shaping the optimization landscape, while Ge et al. (2015) examined the difficulty of escaping saddle points in nonconvex settings. In parallel, substantial progress has been made on error estimation. For example, De Ryck et al. (2024) quantified approximation and optimization errors for physics-informed neural networks, and Shin et al. (2020) established convergence results for PINNs. More recently, Zhao & Luo (2025) proved convergence for a broad class of DNN-based PDE solvers for nonlinear PDEs, and Jiao et al. (2025) presented a comprehensive error analysis of overparameterized three-layer networks trained with projected gradient descent in the deep Ritz method. Most of these results study error through the lenses of generalization and optimization, emphasizing approximation capacity and the optimization gap.

## 1.2 OUR CONTRIBUTION

In this work, we present a continuous-time framework for analyzing the local training dynamics of a stochastic gradient descent variant, termed noisy regularized SGD (see later sections), in deep neural network-based PDE solvers. Our main contributions are summarized below:

(i) **Unified SDE-based continuous-time modeling.** We develop an SDE approximation for noisy regularized SGD in DNN-based PDE solvers and derive rigorous weak-error bounds that precisely quantify the discrepancy between the discrete algorithm and its continuous-time model. In particular, we decompose the weak error into contributions from trajectories that remain within a bounded domain and from rare exit events (Theorem 1).

(ii) **New error analysis via invariant measures.** We introduce a new perspective on error estimation by using the invariant measures of the SDE, and derive its asymptotic formulation (Proposition 5).

(iii) **The impact of stochasticity on stability and solution accuracy.** Through experiments, we systematically study how stochasticity affects the stable step-size regime and constrains solution accuracy even with stable step sizes (Section 4).

**Organization of the paper.** The remainder of this paper is organized as follows. Section 2 reviews the necessary background, notation, and preliminaries. Section 3 introduces the noisy regularized SGD and its associated SDE, derives weak-error bounds, and offers a new perspective on error estimation. Section 4 presents numerical experiments illustrating the impact of stochasticity. Section 5 concludes and outlines future directions. Detailed proofs and additional materials supporting the main text and experiments are provided in the appendix.

## 2 PRELIMINARIES

In this section, we introduce the mathematical framework for our study. We briefly review the use of physics-informed neural networks for solving elliptic equations, describe the stochastic gradient descent algorithm for training, and present an intuitive continuous approximation of SGD. Key assumptions and notations are also summarized, laying the groundwork for later analyses.

### 2.1 PROBLEM SETUP AND BACKGROUND

We focus on physics-informed neural networks as a representative class of DNN-based PDE solvers, which embed physical laws and boundary conditions into the loss. This approach offers insights into DNN-based PDE solvers while highlighting challenges unique to learning with physical constraints. For further clarity and simplicity, we consider second-order elliptic PDEs and use two-layer neural networks in our analysis. This setting captures the essential ideas without unnecessary complexity, and our results readily extend to more general equations and deeper network architectures.

As previously discussed, the primary focus of this work is a class of second-order elliptic partial differential equation of the form:

$$\mathcal{L}u = f \quad \text{in} \quad U \subseteq \mathbb{R}^d, \tag{1}$$

where $\mathcal{L}$ denotes a second-order elliptic operator defined as $\mathcal{L} = -\sum_{i,j=1}^{d} a^{ij}(\boldsymbol{x})\, \partial_{x_i x_j} + \sum_{i=1}^{d} b^i(\boldsymbol{x})\, \partial_{x_i} + c(\boldsymbol{x})$, with $x_i$ being the $i$-th component of $\boldsymbol{x}$. Here, the functions $a^{ij}, b^i, c, f : U \to \mathbb{R}$ are prescribed. For clarity, we assume $U$ is a bounded open subset of $\mathbb{R}^d$ with unit volume. To approximate the solution of Eq.(1), we employ a two-layer neural network with width $m$, described by

$$u_{\boldsymbol{\theta}}(\boldsymbol{x}) = u(\boldsymbol{x}; \boldsymbol{\theta}) = a_0 + \sum_{k=1}^{m} a_k\, \sigma(\boldsymbol{w}_k^{\mathsf{T}} \boldsymbol{x} + b_k), \tag{2}$$

where $\boldsymbol{w}_k \in \mathbb{R}^d$, $a_0, a_k, b_k \in \mathbb{R}$, and $\sigma$ is an activation function. The parameter vector is given by $\boldsymbol{\theta} = \text{vec}\{a_0, \{a_k, \boldsymbol{w}_k, b_k\}_{k=1}^m\} \in \mathbb{R}^M$, where $M = 1 + m(d+2)$ is the total number of parameters.

Within the PINN framework, the approximate solution to Eq.(1) is obtained by minimizing the loss

$$L(\boldsymbol{\theta}) = \int_U \left(\mathcal{L}u_{\boldsymbol{\theta}}(\boldsymbol{x}) - f(\boldsymbol{x})\right)^2 \mathrm{d}\boldsymbol{x}. \tag{3}$$

To optimize this loss, a widely used algorithm is stochastic gradient descent (SGD), which estimates the gradient at each iteration from a mini-batch of $n$ points randomly sampled from $U$. We define the pointwise loss $\ell : \mathbb{R}^M \times U \to \mathbb{R}$ by $\ell(\boldsymbol{\theta}, \boldsymbol{x}) = (\mathcal{L}u_{\boldsymbol{\theta}}(\boldsymbol{x}) - f(\boldsymbol{x}))^2$. At iteration $k$, let $B_k = (\boldsymbol{x}_i^{(k)})_{i=1}^n$ denote the mini-batch, where $\boldsymbol{x}_i^{(k)}$ are sampled independently and uniformly from $U$. The parameter update is then

$$\boldsymbol{\theta}(k+1) = \boldsymbol{\theta}(k) - \frac{\eta}{n} \sum_{i=1}^{n} \nabla_{\boldsymbol{\theta}} \ell\left(\boldsymbol{\theta}(k), \boldsymbol{x}_i^{(k)}\right), \tag{4}$$

where $\eta$ is the learning rate, and $n$ is the batch size. The sequence $\{\boldsymbol{\theta}(k)\}$ thus forms a discrete-time stochastic process determined by the sequence of randomly sampled mini-batches.

**Remark 1** (Boundary conditions). *Our formulation of the PDE* (1) *and the loss* (3) *does not include boundary terms. For Dirichlet conditions, boundary enforcement reduces to supervised learning on $\partial U$, for which continuous-time modeling of SGD is well studied in Li et al. (2017; 2019). We omit the boundary loss to highlight the distinct features and challenges of the PINN framework.*

**Remark 2** (Empirical loss function). *The loss Eq.(3) is not the empirical loss used in practice. Nonetheless, all results extend directly to the empirical setting, including cases with training data drawn from specific distributions. This choice streamlines the presentation.*

### 2.2 AN INTUITIVE CONTINUOUS MODEL FOR SGD

Stochastic fluctuations from discrete SGD updates significantly affect convergence and stability. Continuous-time modeling provides a useful tool to analyze these effects. Below, we present an informal continuous approximation of SGD via stochastic modified equations (Li et al., 2017; 2019).

Specifically, we rewrite Eq.(4) as

$$\boldsymbol{\theta}(k+1) = \boldsymbol{\theta}(k) - \eta \nabla L(\boldsymbol{\theta}(k)) + \sqrt{\eta}\, \boldsymbol{V}(k), \tag{5}$$

where $\boldsymbol{V}(k) = \sqrt{\eta}\Big(\nabla L(\boldsymbol{\theta}(k)) - \frac{1}{n}\sum_{i=1}^{n} \nabla_{\boldsymbol{\theta}} \ell(\boldsymbol{\theta}(k), \boldsymbol{x}_i^{(k)})\Big)$. Given $\boldsymbol{\theta}(k)$, a continuous-time approximation of the transition from $k$ to $k+1$ hinges on the first and second moments of $\boldsymbol{V}(k)$, which determine the drift and diffusion coefficients of the associated SDE. We therefore state the following lemma, with proof in Section B.

**Lemma 1** (Conditional gradient noise). *Conditioned on $\boldsymbol{\theta}(k)$, the random vector $\boldsymbol{V}(k)$ has mean zero and its conditional covariance matrix is given by $\mathrm{cov}\,[\boldsymbol{V}(k), \boldsymbol{V}(k)|\boldsymbol{\theta}(k)] = \eta\boldsymbol{\Sigma}(\boldsymbol{\theta}(k))$, where $\boldsymbol{\Sigma}(\boldsymbol{\theta}(k)) = (\Sigma_{ij}(\boldsymbol{\theta}(k)))_{1 \le i,j \le M}$. For each entry, we have*

$$\Sigma_{ij}(\boldsymbol{\theta}) = \frac{1}{n}\left(\int_U \frac{\partial \ell\,(\boldsymbol{\theta}, \boldsymbol{x})}{\partial \theta_i} \frac{\partial \ell\,(\boldsymbol{\theta}, \boldsymbol{x})}{\partial \theta_j}\, \mathrm{d}\boldsymbol{x} - \int_U \frac{\partial \ell\,(\boldsymbol{\theta}, \boldsymbol{x})}{\partial \theta_i}\, \mathrm{d}\boldsymbol{x} \int_U \frac{\partial \ell\,(\boldsymbol{\theta}, \boldsymbol{x})}{\partial \theta_j}\, \mathrm{d}\boldsymbol{x}\right).$$

Based on Lemma 1, we can intuitively derive a continuous-time approximation of SGD using the framework of stochastic modified equations. Consider the following time-homogeneous Itô stochastic differential equation (SDE):

$$\mathrm{d}\boldsymbol{\Theta}_t = \boldsymbol{b}(\boldsymbol{\Theta}_t)\mathrm{d}t + \eta^{\frac{1}{2}} \boldsymbol{\sigma}(\boldsymbol{\Theta}_t)\mathrm{d}\boldsymbol{W}(t), \tag{6}$$

where $\boldsymbol{\Theta}_t \in \mathbb{R}^M$ for $t \ge 0$, and $\boldsymbol{W}(t)$ denotes a standard $M$-dimensional Wiener process. Applying the Euler discretization with step size $\eta$ to Eq.(6), and denoting the discrete-time approximation by $\hat{\boldsymbol{\Theta}}(k)$, we obtain the following iteration:

$$\hat{\boldsymbol{\Theta}}(k+1) = \hat{\boldsymbol{\Theta}}(k) + \eta\, \boldsymbol{b}\big(\hat{\boldsymbol{\Theta}}(k)\big) + \eta^{\frac{1}{2}} \boldsymbol{\sigma}\big(\hat{\boldsymbol{\Theta}}(k)\big)\boldsymbol{Z}_k, \tag{7}$$

where $\boldsymbol{Z}_k$ are i.i.d. standard Gaussian random vectors in $\mathbb{R}^M$. Comparing Eq.(7) with Eq.(5), we see that by choosing $\boldsymbol{b} = -\nabla L$, $\boldsymbol{\sigma} = \boldsymbol{\Sigma}^{1/2}$, the first and second conditional moments are matched. This observation motivates the following intuitive SDE as an approximation for the dynamics of SGD:

$$\begin{cases} \mathrm{d}\boldsymbol{\Theta}_t = -\nabla L(\boldsymbol{\Theta}_t)\mathrm{d}t + \big(\eta\boldsymbol{\Sigma}(\boldsymbol{\Theta}_t)\big)^{1/2}\mathrm{d}\boldsymbol{W}(t), \\ \boldsymbol{\Theta}_0 = \boldsymbol{\theta}(0). \end{cases} \tag{8}$$

**Remark 3.** *$\eta\boldsymbol{\Sigma}(\boldsymbol{\theta}(k))$ is the conditional covariance matrix of $\boldsymbol{V}(k)$ given $\boldsymbol{\theta}(k)$. So $\boldsymbol{\Sigma}$ is a symmetric positive semidefinite matrix. Consequently, there exists a unique real symmetric positive semidefinite matrix $\boldsymbol{\sigma}(\boldsymbol{\theta})$ such that $\boldsymbol{\sigma}(\boldsymbol{\theta})^2 = \boldsymbol{\Sigma}(\boldsymbol{\theta})$. In this sense, the square root of $\boldsymbol{\Sigma}$ is well-defined.*

While the SDE (8) intuitively approximates SGD (4), its coefficients fail to satisfy global Lipschitz continuity and linear growth, so classical SDE theory does not guarantee well-posedness. In the next section, we address this by introducing a variant of SGD.

## 2.3 Assumptions and notations

To ensure a well-posed SDE model, we impose the following assumptions on the coefficients in Eq.(1) and the neural network activations:

**Assumption 1** (Coefficient regularity). $a^{ij}(\boldsymbol{x})$, $b^i(\boldsymbol{x})$, $c(\boldsymbol{x})$, $f(\boldsymbol{x}) \in L^\infty(U)$, $\quad \forall i, j = 1, \ldots, d$.

**Assumption 2** (Bounded activations). $\sigma$, $\sigma'$, $\sigma'' \in L^\infty(\mathbb{R})$.

These conditions guarantee polynomial bounds for $L(\boldsymbol{\theta})$ (see Lemma 3) and ensure stability of both the PDE solution and the neural network dynamics.

Next, we introduce some notations used throughout this paper. Let $|\cdot|$ denote the vector $L^2$ norm. For a vector $\boldsymbol{v}$, $v_i$ denotes its $i$-th entry. For a matrix $\boldsymbol{A}$, $\|\boldsymbol{A}\|_{\mathrm{F}}$ denotes the Frobenius norm. Let $B_R$ be the ball of radius $R$ centered at the origin in $\mathbb{R}^M$. Furthermore, throughout this paper, we use boldface letters to represent vectors or matrices.

## 3   FOR THE NOISY REGULARIZED SGD

Revisiting Eq.(8), two issues arise. First, the drift is not of linear growth, and no explicit upper bound on its growth is available. Second, the noise driven by $\boldsymbol{\sigma}$ may be degenerate, since the covariance $\boldsymbol{\Sigma}$ can be only semidefinite rather than positive definite. To control the drift (i.e., the gradient of the loss) and prevent gradient explosion, a common technique is implicit regularization, which augments the loss with a regularization term to stabilize the dynamics. Accordingly, we focus on a noisy, regularized variant of SGD.

Specifically, for any fixed integer $s \geq 10$, we consider a regularized loss given by $L_\delta(\boldsymbol{\theta}) = L(\boldsymbol{\theta}) + \delta|\boldsymbol{\theta}|^{2s}$. The additional term controls the growth of the original loss and can aid in establishing well-posedness for the continuous stochastic dynamics. At each iteration, we further inject independent noise $\boldsymbol{g}(k)$. The resulting noisy regularized SGD updates the parameters as

$$\boldsymbol{\theta}(k+1) = \boldsymbol{\theta}(k) - \frac{\eta}{n}\sum_{i=1}^{n}\nabla_{\boldsymbol{\theta}}\ell_\delta\left(\boldsymbol{\theta}(k), \boldsymbol{x}_i^{(k)}\right) + \sqrt{\varepsilon}\boldsymbol{g}(k), \tag{9}$$

where $\ell_\delta(\boldsymbol{\theta}, \boldsymbol{x}) = \ell(\boldsymbol{\theta}, \boldsymbol{x}) + \delta|\boldsymbol{\theta}|^{2s}$ and $\boldsymbol{g}(k) \sim N(\boldsymbol{0}, \boldsymbol{I})$. We provide numerical experiments to test this algorithm in Section J. Following the approach in Section 2.2, we can informally derive its continuous-time approximation, the noisy regularized SDE:

$$\mathrm{d}\boldsymbol{\Theta}_t = \boldsymbol{b}_\delta(\boldsymbol{\Theta}_t)\mathrm{d}t + \eta^{\frac{1}{2}}\boldsymbol{\sigma}_\varepsilon(\boldsymbol{\Theta}_t)\mathrm{d}\boldsymbol{W}(t), \tag{10}$$

where $\boldsymbol{b}_\delta(\boldsymbol{\theta}) = -\nabla L_\delta(\boldsymbol{\theta})$ and $\boldsymbol{\sigma}_\varepsilon(\boldsymbol{\theta}) = [\boldsymbol{\Sigma}(\boldsymbol{\theta}) + \varepsilon\boldsymbol{I}]^{\frac{1}{2}}$.

**Remark 4** (Choice of regularization term). *In our setup, we add the regularization term $|\boldsymbol{\theta}|^{2s}$ ($s \geq 10$), which may appear unusual compared to the more common $|\boldsymbol{\theta}|^2$ used in practice. However, this higher-order term is necessary to ensure theoretical rigor, as powers lower than $|\boldsymbol{\theta}|^{20}$ are insufficient to control the potential unbounded growth of the loss function at infinity (see Lemma 3).*

The noisy regularized SDE remedies the shortcomings of Eq.(8). In Section C, we establish existence and uniqueness of solution to Eq.(10) via a Lyapunov function $V$ (see Lemma 3) enabled by the regularization term $|\boldsymbol{\theta}|^{2s}$. A natural question is whether the discrete algorithm (9) converges to the continuous model (10). Li et al. (2019) proved weak convergence of SGD to its limiting SDE under a global Lipschitz condition in supervised learning. In our setting, however, the coefficients of Eq.(10) are only locally Lipschitz, so convergence can be established only on bounded domains. Below, we provide the definitions and results for local weak convergence.

**Definition 1** (Polynomial growth functions). *Let $G$ denote the set of continuous functions $\mathbb{R}^M \to \mathbb{R}$ with at most polynomial growth, i.e., $g \in G$ if there exist positive integers $\kappa_1, \kappa_2 > 0$ such that*

$$|g(\boldsymbol{\theta})| \leq \kappa_1(1 + |\boldsymbol{\theta}|^{\kappa_2}), \text{ for all } \boldsymbol{\theta} \in \mathbb{R}^M.$$

*Moreover, for integer $\alpha \geq 1$ we denote by $G^\alpha$ the set of $\alpha$-times continuously differentiable functions, which together with its partial derivatives up to and including order $\alpha$, belong to $G$.*

**Definition 2** (Stopping times). *Let $\boldsymbol{\Theta}^{(\eta)}(t)$ denote the solution to Eq.(10) with $\boldsymbol{\Theta}^{(\eta)}(0) = \boldsymbol{\theta}$, and let $\{\boldsymbol{\theta}^{(\eta)}(k)\}$ be the sequence generated by Eq.(9) with learning rate $\eta > 0$ and initial value $\boldsymbol{\theta}^{(\eta)}(0) = \boldsymbol{\theta}$. Define the exit time of $\boldsymbol{\Theta}^{(\eta)}(t)$ from $B_R$ as $\tau_R^{(\eta)}(\boldsymbol{\theta}) = \inf\{t \geq 0: \boldsymbol{\Theta}^{(\eta)}(t) \notin B_R\}$.*

*The piecewise-constant interpolation of the discrete iterates is denoted by*

$$\bar{\boldsymbol{\theta}}^{(\eta)}(t) := \boldsymbol{\theta}^{(\eta)}(k), \qquad t \in [k\eta, (k+1)\eta), \ k \in \mathbb{N}_0.$$

*Define the exit time of the interpolated process from $B_R$ as $\Gamma_R^{(\eta)}(\boldsymbol{\theta}) = \inf\{t \geq 0: \bar{\boldsymbol{\theta}}^{(\eta)}(t) \notin B_R\}$.*

**Proposition 1** (Local weak convergence). *Fix any $T > 0$. Let $\boldsymbol{\Theta}^{(\eta)}(t)$ denote the solution to the SDE (10) with initial condition $\boldsymbol{\Theta}(0) = \boldsymbol{\theta}$, and let $\boldsymbol{\theta}^{(\eta)}(k)$ be the $k$-th iterate of the discrete model (9) with the same initial value $\boldsymbol{\theta}(0) = \boldsymbol{\theta}$ and step size $\eta$. Then for any $h \in G^6$, $R > 0$, and $\boldsymbol{\theta} \in \mathbb{R}^M$, there exists a constant $C(R, T) > 0$ independent of $\eta$ such that for all $0 < \eta < 1$ and $k = 0, 1, ..., \lfloor\frac{T}{\eta}\rfloor$, we have*

$$\left|\mathbb{E}\left[\left(h\left(\boldsymbol{\Theta}^{(\eta)}(k\eta)\right) - h\left(\boldsymbol{\theta}^{(\eta)}(k)\right)\right)\mathbb{1}_{\left\{k\eta \leq \tau_R^{(\eta)}(\boldsymbol{\theta})\right\}}\mathbb{1}_{\left\{k\eta \leq \Gamma_R^{(\eta)}(\boldsymbol{\theta})\right\}}\right]\right| \leq C(R, T)\eta.$$

**Remark 5.** *We work on a product probability space that carries both the continuous-time noise (driving the SDE) and the discrete-time randomness (from the SGD updates). In Section D, we detail the construction of this product space, clarify the notion of weak convergence used here (including the role of the stopping times).*

The above proposition, proved in Section D.3, establishes local weak convergence of order 1. To complete the analysis, it remains to quantify the probability that the trajectories of the noisy regularized SGD and the SDE stay within $B_R$.

**Proposition 2** (Bounded exit time for SDE)**.** *There exists a constant $c$ independent of $\eta$ such that for all $T > 0$ and initial point $\boldsymbol{\theta} \in \mathbb{R}^M$, $\mathbb{P}\left(\tau_R^{(\eta)}(\boldsymbol{\theta}) \leq T\right) \leq \frac{\mathrm{e}^{cT} V(\boldsymbol{\theta})}{\inf_{|\boldsymbol{\xi}| \geq R} V(\boldsymbol{\xi})} \to 0 \quad as \quad R \to \infty$, where $V = L_\delta + 1$ is the Lyapunov function defined in Lemma 3.*

The previous proposition, proved in Section F.1, shows that for any fixed $T$, the SDE trajectory remains in $B_R$ with high probability when $R$ is large enough. For the noisy regularized SGD, we aim for an analogous high-probability bound that holds uniformly over all step sizes $\eta$. Because the SGD iterates form a family of processes indexed by $\eta$, such uniform estimates require moment bounds that are independent of $\eta$. Establishing these bounds is generally nontrivial and remains open in many settings; therefore, we adopt the following standard assumption to enable our analysis.

**Assumption 3** (Uniform boundedness of discrete dynamics)**.** *Fix any $T > 0$. Let $\boldsymbol{\theta}^{(\eta)}(k)$ denote the $k$-th iterate of the discrete model* (9) *with step size $\eta$. We assume that there exist a positive integer $p$ and a constant $C_{p,T}$, independent of $\eta$, such that $\sup_{0 < \eta < 1} \mathbb{E} \sup_{0 \leq k \leq \lfloor T/\eta \rfloor} \left|\boldsymbol{\theta}^{(\eta)}(k)\right|^{2p} \leq C_{p,T}$.*

We provide further explanation of this assumption, as well as empirical justification, in the appendix Section E. Based on Assumption 3, we can estimate the probability that $\bar{\boldsymbol{\theta}}^{(\eta)}(t)$ exits a given ball within time $T$ directly.

**Proposition 3.** *Under Assumption 3, we have $\mathbb{P}\left(\Gamma_R^{(\eta)}(\boldsymbol{\theta}) \leq T\right) \leq \frac{C_{p,T}}{R^{2p}} \to 0$ as $R \to \infty$, for any $0 < \eta < 1$.*

This proposition shows that the probability of the process $\bar{\boldsymbol{\theta}}^{(\eta)}(t)$ exiting the ball $B_R$ within time $T$ decays polynomially as $R \to \infty$. Combining Proposition 1, Proposition 2, and Proposition 3, we are now in a position to state the main result of this section.

**Theorem 1** (Approximate order-1 weak convergence)**.** *Fix $T > 0$. Assume that Assumptions 1 to 3 hold. Then for any bounded function $h \in G^6$, any $R > 0$, and $\boldsymbol{\theta} \in \mathbb{R}^M$, there exist positive constants $C(R,T)$ independent of $\eta$ such that for all $k = 0, 1, \ldots, \lfloor T/\eta \rfloor$, we have*

$$\left| \mathbb{E}\, h\left(\boldsymbol{\Theta}^{(\eta)}(k\eta)\right) - \mathbb{E}\, h\left(\boldsymbol{\theta}^{(\eta)}(k)\right) \right| \leq C(R,T)\eta + 2\|h\|_\infty \left( \frac{e^{cT} V(\boldsymbol{\theta})}{\inf_{|\boldsymbol{\xi}| \geq R} V(\boldsymbol{\xi})} + \frac{C_{p,T}}{R^{2p}} \right),$$

*where $c$, $V$, and $C_{p,T}$ are as defined in Proposition 2, Lemma 3, and Assumption 3, respectively.*

This theorem (see Section F.2 for the proof) establishes an approximate order-1 local weak convergence between the noisy regularized SGD and its continuous SDE counterpart. The error bound has two parts: the term $C(R,T)\eta$ captures local weak convergence inside $B_R$, while the second term is the probability that the processes exit $B_R$. By taking $R$ large, the latter can be made arbitrarily small, though $C(R,T)$ may grow accordingly. This is possibly the best that can be achieved without any additional assumptions. Moreover, Section G collects several properties of the SDE solution, offering further insight into SGD behavior for sufficiently small step sizes.

### 3.1 A NEW UNDERSTANDING OF ERROR ESTIMATES

In this subsection, we take a different view of error estimation. Over sufficiently long horizons, the SDE solution converges to its stationary distribution, so the long-term error can be characterized by the expected loss under this distribution. For small step sizes, this provides an approximate description of the long-run behavior of noisy regularized SGD. We begin by establishing existence and uniqueness of the SDE's invariant measure.

**Proposition 4** (Unique invariant measure)**.** *SDE* (10) *has a unique invariant measure $\mu$, and it is ergodic. Moreover, the distribution $\mu$ has a density $p(\theta)$ with respect to Lebesgue measure in $\mathbb{R}^M$.*

*This density is the unique bounded solution of the equation*

$$\mathcal{A}^* p = \frac{\eta}{2} \sum_{i,j=1}^{M} \frac{\partial^2}{\partial \theta_i \partial \theta_j} \left( (\Sigma_\varepsilon)_{ij} \, p \right) - \sum_{i=1}^{M} \frac{\partial}{\partial \theta_i} \left( b_{\delta,i} p \right) = 0,$$

*satisfying the additional condition $\int_{\mathbb{R}^M} p(\boldsymbol{\theta}) \, d\boldsymbol{\theta} = 1$.*

Using the WKB approximation in Section H, we obtain $p(\boldsymbol{\theta}) \sim \exp\left( \frac{1}{\beta} S_0(\boldsymbol{\theta}) \right)$. We estimate the error $\int_{\mathbb{R}^M} L(\boldsymbol{\theta}) \, p(\boldsymbol{\theta}) \, d\boldsymbol{\theta}$ using the Laplace method. To this end, we assume $S_0 \in C^2$, which also ensures the same regularity for $p$. The following lemma shows that $S_0$ has only finitely many maximizers, thereby justifying the application of the Laplace method.

**Lemma 2** (Finitely many maximizers). *There are only finitely many maximum points of $S_0$ in $\mathbb{R}^M$.*

For completeness, the proof of Lemma 2 and a brief introduction to the Laplace method are both included in Section I. Applying the Laplace method, we directly obtain:

**Proposition 5** (Error estimates). *Assume $\{\boldsymbol{\theta}^j\}_{j \in J}$ is the set of maximizers of $S_0$. Then the following asymptotic approximation holds for the error:*

$$\int_{\mathbb{R}^M} L(\boldsymbol{\theta}) p(\boldsymbol{\theta}) \, d\boldsymbol{\theta} \sim (2\pi\beta)^{\frac{M}{2}} \sum_{j \in J} \frac{L(\boldsymbol{\theta}^j) e^{\frac{1}{\beta} S_0(\boldsymbol{\theta}^j)}}{\det \left( \nabla^2 S_0 \left( \boldsymbol{\theta}^j \right) \right)^{1/2}}, \quad as \quad \beta \to 0,$$

*where $\beta = \frac{\eta}{n}$.*

**Remark 6** (Unique maximizer and gradient-flow consistency). *If $S_0$ has a unique and nondegenerate maximizer $\theta^\star$, the Laplace method can be applied to both the numerator and denominator in $\int_{\mathbb{R}^M} L(\boldsymbol{\theta}) \, p(\boldsymbol{\theta}) \, d\boldsymbol{\theta} \approx \frac{\int_{\mathbb{R}^M} L(\boldsymbol{\theta}) \, e^{S_0(\boldsymbol{\theta})/\beta} \, d\boldsymbol{\theta}}{\int_{\mathbb{R}^M} e^{S_0(\boldsymbol{\theta})/\beta} \, d\boldsymbol{\theta}}$, which yields the asymptotic limit $\int_{\mathbb{R}^M} L(\boldsymbol{\theta}) \, p(\boldsymbol{\theta}) \, d\boldsymbol{\theta} \sim L(\theta^\star), \quad \beta \to 0$. Since $\beta = \eta/n$, the limit $\beta \to 0$ corresponds to $\eta \to 0$ and $n \to \infty$ (the full-batch regime). In this case, the noisy dynamics converge to the gradient flow, and the expected loss concentrates at the equilibrium, consistent with the previous discussion.*

The proposition characterizes the leading-order behavior of the long-run error $\int_{\mathbb{R}^M} L(\theta) \, p(\theta) \, d\theta$ via the Laplace method: the dominant contributions come from the maximizers of $S_0$, weighted by the local quadratic geometry through $|\nabla^2 S_0(\theta^j)|^{-1/2}$. In the small-$\beta$ regime (small step size and near full-batch sampling), the error is thus governed by a finite sum over these modes; in the special case of a unique maximizer, it reduces to the loss evaluated at that point, consistent with the gradient-flow.

**Remark 7** (On the generality of the SDE framework). *The core technical contribution of this work is establishing the well-posedness of the limiting SDE for non-globally Lipschitz loss landscapes, a common scenario with deep neural networks. This is achieved by constructing a Lyapunov function (the regularized loss itself) to control the growth of the SDE dynamics (as in Lemma 3). Consequently, our analytical framework is not limited to the specific PDEs, network architectures, or losses presented in this section. Its applicability extends to any setting where a suitable Lyapunov function can be constructed to control the corresponding stochastic dynamics, thereby enabling weak error estimates for a broad class of problems in scientific machine learning.*

## 4 EXPERIMENTS: IMPACT OF STOCHASTICITY ON SGD PERFORMANCE

To investigate the effect of stochasticity, we consider the second-order ODE $u''(x) = f(x)$, $x \in [-1, 1]$, with exact solution $u(x) = \tanh(2x + 1)$. The source term $f(x)$ and boundary conditions are set accordingly. We use a two-layer neural network of width 10 in the PINN framework: $u(x; \boldsymbol{\theta}) = \sum_{k=1}^{10} a_k \tanh(w_k x + b_k)$. The neural network can exactly represent the solution, enabling an explicit characterization of all global minimizers. The training set comprises 1,000 uniformly sampled points in $[-1, 1]$. To isolate the effect of stochasticity, we compare SGD and GD under identical settings. We examine two regimes: (i) near a global minimizer with low sharpness and (ii) with high sharpness. The results indicate that the relative advantages of SGD versus GD differ across these regimes.

## 4.1 REGIME 1: NEAR A GLOBAL MINIMIZER WITH SMALL SHARPNESS

In this experiment, we evaluate SGD and GD near a global minimizer $\boldsymbol{\theta}^*$, with $a_0 = 1$, $w_0 = 2$, $b_0 = 1$ for the first neuron and other parameters zero. The sharpness at this point is about $31.14363$, giving an approximate critical stable learning rate $\eta^{**} = 0.06422 \ (2/sharpness)$ for GD.

**Experiment 1: Stability domains of learning rates for SGD and GD.** We performed 50 random initializations near $\boldsymbol{\theta}^*$. For each initialization, both SGD and GD were run for 300 steps across 50 increasing step sizes. We recorded the loss at step 300 for each algorithm under all step sizes. The results show that, for all 50 initializations, whenever the step size slightly exceeds $\eta^{**}$, the loss at step 300 for SGD exhibits a sharp increase with respect to smaller step sizes, unlike GD, which remains stable across all step sizes. An example for one initialization is shown in Figure 1. Detailed experimental settings and additional results are provided in Section K.1.1. These results demonstrate the much narrower stability range of learning rates for SGD compared to GD near global minimizer $\boldsymbol{\theta}^*$ with small sharpness.

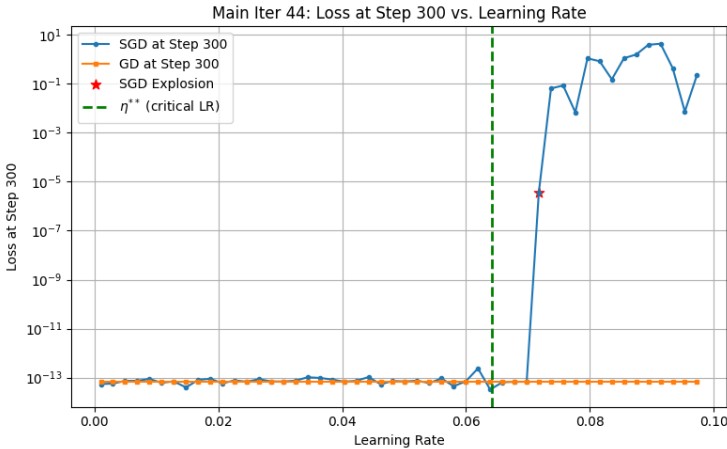

Figure 1: One typical trial comparing the stability domains of SGD and GD near $\boldsymbol{\theta}^*$.

**Experiment 2: Effect of stochasticity on solution accuracy within the stable learning rate regime.** We study the effect of stochasticity using five learning rates: 0.001, 0.002, 0.003, 0.004, and 0.005 (all $< \eta^{**}$). For each rate, we run SGD 50 times (batch size 32, 3000 iterations) with random initializations near $\boldsymbol{\theta}^*$. The mean prediction is evaluated on 10,000 test points, and relative $L^2$ errors are reported in Table 1. For step sizes below 0.003, performance remains consistent. At 0.004 and 0.005, noise effects become pronounced and the averaged solution accuracy drops sharply. In contrast, 50 runs of GD with $\eta = 0.005$ yield a relative $L^2$ error almost an order of magnitude smaller than SGD, underscoring the detrimental effect of stochasticity. Further details and an analysis of step size effects on solution variance are in Section K.1.2. Overall, these results show that greater stochasticity (larger step size) significantly degrades SGD performance.

Table 1: Relative $L^2$ error of the mean function at different learning rates.

| Learning Rate $\eta$ | 0.001 | 0.002 | 0.003 | 0.004 | 0.005 | 0.005(GD) |
|---|---|---|---|---|---|---|
| Relative $L^2$ Error | 3.60e-07 | 3.57e-07 | 3.53e-07 | 2.74e-05 | 3.51e-05 | 7.45e-06 |

**Experiment 3: Mechanism of SGD blow-up at large step sizes.** To investigate the mechanism behind the blow-up of SGD at large learning rates, we train the network (initialized near $\boldsymbol{\theta}^*$) under $\eta = 0.096327$. Both SGD and GD are run for 3500 steps, recording the loss at each iteration. For SGD, we also monitor the Frobenius norm of the covariance matrix $\eta\Sigma$. Results are shown in Figure 2. With this large step size, GD remains stable, while SGD quickly diverges. The loss curve

for SGD closely follows that of the covariance norm, indicating that increased stochastic fluctuations drive the instability and blow-up of SGD at large learning rates.

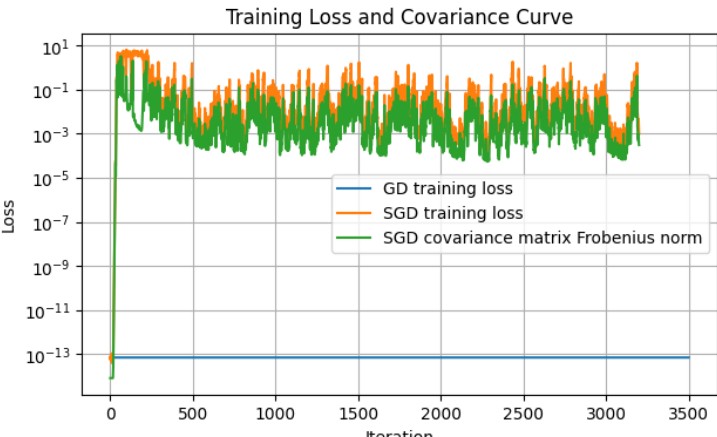

Figure 2: Evolution of the loss and Frobenius norm of the covariance matrix with $\eta = 0.096327$.

### 4.2 REGIME 2: NEAR A GLOBAL MINIMIZER WITH LARGE SHARPNESS

Before presenting our experiments in this regime, we briefly explain why parameter vectors with very large sharpness can arise in practice. The *edge of stability* phenomenon (Cohen et al., 2021) shows that optimization often drives parameters to regions where the sharpness approaches $2/\eta$, with $\eta$ the learning rate. At this time, the loss begins to oscillate, but reducing $\eta$ restores stability. By repeatedly decreasing the step size at the edge of stability, the optimizer can reach global minimizers with very large sharpness, explaining why such regions can be encountered.

We next present our experiments. We select a global minimizer $\theta^{**}$ with sharpness about $1.14 \times 10^5$, giving a theoretical GD stability threshold of approximately $1.75 \times 10^{-5}$ (see Table 2). Across 50 trials, parameters are randomly initialized near $\theta^{**}$, and the critical GD step size is computed for each initialization. Both SGD and GD are run for 600 steps at this critical step size plus $1 \times 10^{-6}$, and their loss trajectories are recorded. In all 50 trials, GD causes the loss to surge from $10^{-6}$ to $10^2$ within 100 steps, then slowly declines and stabilizes around $10^{-4}$. By contrast, SGD steadily reduces the loss from $10^{-6}$ to $10^{-10}$. Two representative runs are shown in Figure 3, illustrating this qualitative difference. These results indicate that, near this highly sharp minimum, even a slight increase beyond the stability threshold causes GD to diverge, while SGD continues to make progress. Notably, this behavior contradicts the theoretical predictions of Wu et al. (2018), and understanding its cause remains an open question for future work. Further experimental details are given in Section K.2.

Our above experiments employ an ordinary differential equation (ODE) perspective to interpret the effect of stochastic mini-batch noise in SGD, as compared to deterministic gradient descent, on the solution behavior of the corresponding ODE. To further support the generality of our conclusion, we present additional experiments in the appendix Section K.3 for both the Helmholtz equation and the Allen–Cahn equation, following the same rationale and experimental protocol. The results consistently demonstrate that similar observations hold in these PDE settings as well, thereby reinforcing the practical validity of our conclusions across a wider range of problems.

## 5 CONCLUSION AND DISCUSSION

In this work, we present a rigorous analysis of noisy, regularized SGD for DNN-based PDE solvers. By introducing an SDE approximation, we derive precise weak error estimates between the discrete and continuous dynamics. Our general framework applies to diverse solver types, a broad range of PDEs, and various network architectures, as long as the loss function can be controlled by a suit-

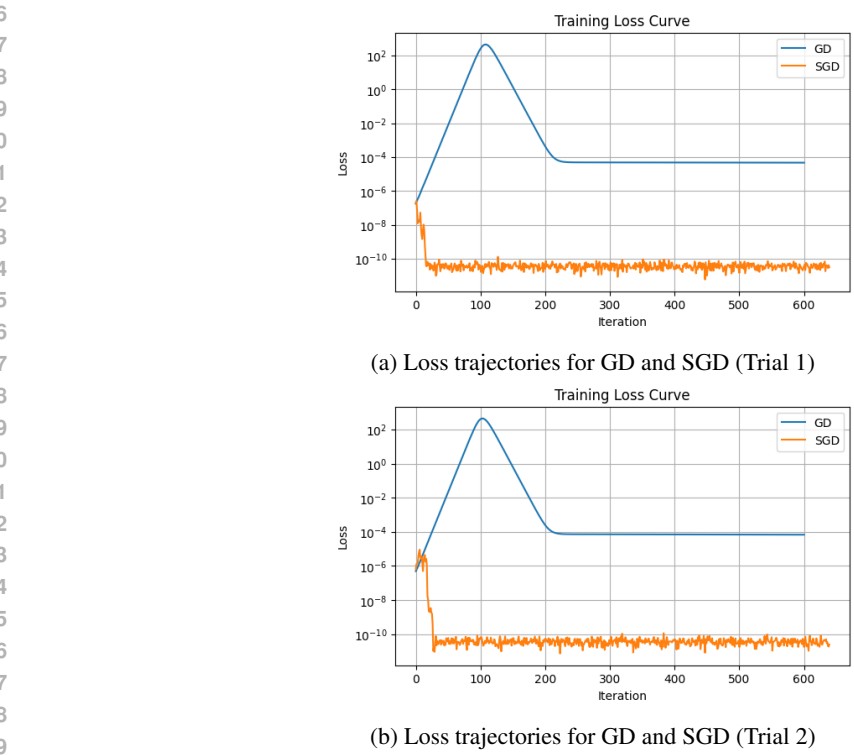

(a) Loss trajectories for GD and SGD (Trial 1)

(b) Loss trajectories for GD and SGD (Trial 2)

Figure 3: Representative loss trajectories of GD and SGD for two random initializations near $\theta^{**}$.

able polynomial. Notably, by relaxing the global Lipschitz condition, our results directly encompass many networks used in scientific computing. Systematic experiments reveal that higher noise increases solution variance and reduces accuracy, and that step size stability for SGD and GD can differ significantly near different global minima.

**Limitations.** This study has several limitations. First, our analysis focuses primarily on SGD and does not extend to other popular optimizers like Adam or L-BFGS, whose dynamics may differ substantially. Second, while we use an SDE to model the training, we do not leverage this SDE framework to proactively guide or improve algorithm design. Finally, our noise model is confined to Brownian motion; exploring broader models, such as SDEs driven by Lévy processes, could more accurately capture the stochasticity in practical training scenarios. Addressing these limitations presents valuable directions for future research.

ETHICS STATEMENT

This paper adheres to the ICLR Code of Ethics, as acknowledged and committed to by all authors.

REPRODUCIBILITY STATEMENT

All theoretical results in this paper are accompanied by complete proofs provided in the appendix. For all experiments presented in both the main text and the appendix, detailed descriptions of the experimental setups are provided in the appendix to ensure reproducibility. Furthermore, we have included all source code (in a compressed archive) as supplementary material to further support the reproducibility of our results.

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

## A  THE USE OF LARGE LANGUAGE MODELS (LLMS)

We used a large language model (GPT-5) to polish the language of the manuscript. Specifically, we drafted the initial versions of all sections ourselves, and then employed GPT-5 to refine the wording and clarity of selected passages—primarily introductory and expository paragraphs. The model did not contribute to research ideation, methodology, experiments, analyses, or conclusions.

## B  PROOF OF LEMMA 1

*Proof of Lemma 1.* Let $x$ be a random variable uniformly distributed on $U \subseteq \mathbb{R}^d$ with respect to the Lebesgue measure. By definition, we have

$$\mathbb{E}[\ell(\boldsymbol{\theta}, \boldsymbol{x})] = \int_U \ell(\boldsymbol{\theta}, \boldsymbol{x}) \, \mathrm{d}\boldsymbol{x} = L(\boldsymbol{\theta}).$$

For each $i = 1, \ldots, M$, we have

$$\frac{\partial L(\boldsymbol{\theta})}{\partial \theta_i} = \frac{\partial}{\partial \theta_i} \mathbb{E}_{\boldsymbol{x}}[\ell(\boldsymbol{\theta}, \boldsymbol{x})] = \mathbb{E}_{\boldsymbol{x}} \left[ \frac{\partial}{\partial \theta_i} \ell(\boldsymbol{\theta}, \boldsymbol{x}) \right],$$

where the interchange of differentiation and expectation is justified by regularity conditions.

Therefore, it follows immediately that

$$\mathbb{E}\left[ \, \boldsymbol{V}(k) \, | \, \boldsymbol{\theta}(k) \, \right] = 0.$$

For the conditional covariance, by definition we have

$$\mathrm{cov}\left[ \, \boldsymbol{V}(k), \boldsymbol{V}(k) | \, \boldsymbol{\theta}(k) \right] = \eta \, \mathbb{E}\Bigg[ \left( \nabla L(\boldsymbol{\theta}(k)) - \frac{1}{n} \sum_{i=1}^{n} \nabla_{\boldsymbol{\theta}} \ell(\boldsymbol{\theta}(k), \boldsymbol{x}_i^{(k)}) \right)$$
$$\times \left( \nabla L(\boldsymbol{\theta}(k)) - \frac{1}{n} \sum_{i=1}^{n} \nabla_{\boldsymbol{\theta}} \ell(\boldsymbol{\theta}(k), \boldsymbol{x}_i^{(k)}) \right)^{\top} \Bigg| \, \boldsymbol{\theta}(k) \Bigg].$$

Define the matrix $\boldsymbol{\Sigma}(\boldsymbol{\theta}(k))$ as the conditional covariance above (without the factor $\eta$), i.e.,

$$\boldsymbol{\Sigma}(\boldsymbol{\theta}(k)) = \mathbb{E}\Bigg[\left(\nabla L(\boldsymbol{\theta}(k)) - \frac{1}{n}\sum_{i=1}^{n}\nabla_{\boldsymbol{\theta}}\ell(\boldsymbol{\theta}(k), \boldsymbol{x}_i^{(k)})\right)$$

$$\times \left(\nabla L(\boldsymbol{\theta}(k)) - \frac{1}{n}\sum_{i=1}^{n}\nabla_{\boldsymbol{\theta}}\ell(\boldsymbol{\theta}(k), \boldsymbol{x}_i^{(k)})\right)^{\top}\Bigg|\ \boldsymbol{\theta}(k)\Bigg].$$

Calculating entry-wise for $i, j = 1, \ldots, M$, using the independence of samples in the mini-batch, we obtain

$$\Sigma_{ij}(\boldsymbol{\theta}(k)) = \frac{1}{n}\left[\int_U \frac{\partial\ell(\boldsymbol{\theta}(k), \boldsymbol{x})}{\partial\theta_i}\frac{\partial\ell(\boldsymbol{\theta}(k), \boldsymbol{x})}{\partial\theta_j}\,\mathrm{d}\boldsymbol{x} - \int_U \frac{\partial\ell(\boldsymbol{\theta}(k), \boldsymbol{x})}{\partial\theta_i}\,\mathrm{d}\boldsymbol{x}\int_U \frac{\partial\ell(\boldsymbol{\theta}(k), \boldsymbol{x})}{\partial\theta_j}\,\mathrm{d}\boldsymbol{x}\right].$$

This completes the proof. $\qquad\square$

## C   WELL-POSEDNESS OF SOLUTIONS TO THE NOISY REGULARIZED SDE

A fundamental step is to establish the well-posedness of Eq.(10). To this end, we construct an appropriate Lyapunov function and invoke classical SDE theory to rigorously prove the existence and uniqueness of solutions.

**Lemma 3** (Lyapunov function). *Let $V(\boldsymbol{\theta}) = L_\delta(\boldsymbol{\theta}) + 1$ and let $\mathcal{A}$ denote the infinitesimal generator associated with the SDE* (10)*, that is,*

$$\mathcal{A} = \sum_{i=1}^{M} b_{\delta,i}\partial_i + \frac{\eta}{2}\sum_{i,j=1}^{M}(\Sigma_\varepsilon)_{ij}\partial_i\partial_j,$$

*where $b_{\delta,i}$ is the $i$-th component of $\boldsymbol{b}_\delta$ and $(\Sigma_\varepsilon)_{ij}$ denotes the $(i,j)$-th entry of the matrix $\boldsymbol{\Sigma}_\varepsilon = \boldsymbol{\Sigma} + \varepsilon\boldsymbol{I}$. Then under Assumptions 1 and 2, there exist a constant $c > 0$ independent of $\eta$ such that*

$$\mathcal{A}V(\boldsymbol{\theta}) \leq cV(\boldsymbol{\theta}), \quad \forall\boldsymbol{\theta} \in \mathbb{R}^M,$$

*and moreover,*

$$V_R := \inf_{|\boldsymbol{\theta}|>R} V(\boldsymbol{\theta}) \to \infty \quad as \quad R \to \infty.$$

*Proof of Lemma 3.* Without loss of generality, we assume $s = 10$.

Although the coefficients of SDE (10) are not globally Lipschitz, they possess a crucial property: both the drift $\boldsymbol{b}_\delta$ and the diffusion matrix $\boldsymbol{\Sigma}_\varepsilon$ can be bounded by polynomials in $\boldsymbol{\theta}$. We first establish this fact.

Consider the special case where the width of the neural network is $m = 1$, i.e.,

$$u_{\boldsymbol{\theta}}(\boldsymbol{x}) = a_0 + a\sigma(\boldsymbol{w}^{\mathsf{T}}\boldsymbol{x} + b).$$

The gradient then becomes

$$\nabla L(\boldsymbol{\theta}) = \int_U 2(\mathcal{L}u_{\boldsymbol{\theta}} - f)\nabla_{\boldsymbol{\theta}}\mathcal{L}u_{\boldsymbol{\theta}}\,\mathrm{d}\boldsymbol{x}$$

$$= 2\int_U (\mathcal{L}u_{\boldsymbol{\theta}} - f)\,(c(\boldsymbol{x}),\ \mathcal{L}\sigma(\boldsymbol{w}^{\mathsf{T}}\boldsymbol{x} + b),\ \mathcal{L}a\sigma'(\boldsymbol{w}^{\mathsf{T}}\boldsymbol{x} + b)\boldsymbol{x},\ \mathcal{L}a\sigma'(\boldsymbol{w}^{\mathsf{T}}\boldsymbol{x} + b))^{\mathsf{T}}\,\mathrm{d}\boldsymbol{x}.$$

In particular,

$$\int_U (\mathcal{L}u_{\boldsymbol{\theta}} - f)c(\boldsymbol{x})\,\mathrm{d}\boldsymbol{x}$$

$$= \int_U \left[-\sum_{i,j=1}^{d} a^{ij}c\sigma''aw_iw_j + \sum_{i=1}^{d} b^ic\sigma'aw_i + c^2a_0 + c^2a\sigma\right]\mathrm{d}\boldsymbol{x} - \int_U f(\boldsymbol{x})c(\boldsymbol{x})\,\mathrm{d}\boldsymbol{x}.$$

Recalling Assumption 1, we obtain the following inequality

$$\left| \int_U (\mathcal{L}u_{\boldsymbol{\theta}} - f) c(\boldsymbol{x}) \, \mathrm{d}\boldsymbol{x} \right| \leq C \left( \sum_{i,j=1}^d |a w_i w_j| + \sum_{i=1}^d |a w_i| + |a_0| + |a| + 1 \right)$$

$$\leq C \left( \sum_{i,j=1}^d (a^2 + 1)(w_i^2 + 1)(w_j^2 + 1) + \sum_{i=1}^d (a^2 + 1)(w_i^2 + 1) + a_0^2 + a^2 + 1 \right),$$

where the constant $C$ may differ line by line. Similar calculations show that each component of $\nabla L$ can be bounded by a polynomial of degree at most 8.

Now, consider the Hessian of $L$:

$$D^2 L = 2 \int_U (\mathcal{L}u_{\boldsymbol{\theta}} - f) H \, \mathrm{d}\boldsymbol{x} + 2 \int_U \nabla_{\boldsymbol{\theta}} \mathcal{L}u_{\boldsymbol{\theta}} (\nabla_{\boldsymbol{\theta}} \mathcal{L}u_{\boldsymbol{\theta}})^{\mathsf{T}} \, \mathrm{d}\boldsymbol{x},$$

where $H = \nabla_{\boldsymbol{\theta}}^2 (\mathcal{L}u_{\boldsymbol{\theta}})$. More explicitly,

$$H = \begin{pmatrix} 0 & 0 & \mathbf{0} & 0 \\ 0 & 0 & \mathcal{L}(\sigma' \boldsymbol{x}^{\mathsf{T}}) & \mathcal{L}(\sigma') \\ \mathbf{0} & \mathcal{L}(\sigma' \boldsymbol{x}) & \mathcal{L}(a\sigma'' \boldsymbol{x}\boldsymbol{x}^{\mathsf{T}}) & \mathcal{L}(a\sigma'' \boldsymbol{x}) \\ 0 & \mathcal{L}(\sigma') & \mathcal{L}(a\sigma'' \boldsymbol{x}^{\mathsf{T}}) & \mathcal{L}(a\sigma'') \end{pmatrix}.$$

Again, each entry of $D^2 L$ can be bounded by a polynomial of degree no more than 16.

For the covariance, for $i, j = 1, \ldots, M$,

$$\Sigma_{ij}(\boldsymbol{\theta}) = \frac{1}{n} \left( \int_U \frac{\partial \ell(\boldsymbol{\theta}, \boldsymbol{x})}{\partial \theta_i} \frac{\partial \ell(\boldsymbol{\theta}, \boldsymbol{x})}{\partial \theta_j} \, \mathrm{d}\boldsymbol{x} - \int_U \frac{\partial \ell(\boldsymbol{\theta}, \boldsymbol{x})}{\partial \theta_i} \, \mathrm{d}\boldsymbol{x} \int_U \frac{\partial \ell(\boldsymbol{\theta}, \boldsymbol{x})}{\partial \theta_j} \, \mathrm{d}\boldsymbol{x} \right)$$

$$= \frac{4}{n} \Big( \int_U (\mathcal{L}u_{\boldsymbol{\theta}} - f)^2 \frac{\partial \mathcal{L}u_{\boldsymbol{\theta}}}{\partial \theta_i} \frac{\partial \mathcal{L}u_{\boldsymbol{\theta}}}{\partial \theta_j} \, \mathrm{d}\boldsymbol{x} - \int_U (\mathcal{L}u_{\boldsymbol{\theta}} - f) \frac{\partial \mathcal{L}u_{\boldsymbol{\theta}}}{\partial \theta_i} \, \mathrm{d}\boldsymbol{x} \int_U (\mathcal{L}u_{\boldsymbol{\theta}} - f) \frac{\partial \mathcal{L}u_{\boldsymbol{\theta}}}{\partial \theta_j} \, \mathrm{d}\boldsymbol{x} \Big).$$

By the same argument as above, Assumption 1 ensures that $\Sigma_{ij}$, and thus $(\Sigma_\varepsilon)_{ij}$, can be bounded by a polynomial of degree at most 16. The generalization to $m > 1$ is straightforward.

Now, we return to the main claim of the lemma. Since $|\boldsymbol{\theta}|^{20}$ is the only highest-degree component in $V(\boldsymbol{\theta})$, we have

$$V_R := \inf_{|\boldsymbol{\theta}| > R} V(\boldsymbol{\theta}) \to \infty \quad \text{as} \quad R \to \infty.$$

By the definition of the generator $\mathcal{A}$, we have

$$\mathcal{A}V = \sum_{i=1}^M b_{\delta,i} \partial_i V + \frac{\eta}{2} \sum_{i,j=1}^M (\Sigma_\varepsilon)_{ij} \partial_{ij} V$$

$$= -\langle \nabla L_\delta, \nabla L_\delta \rangle + \frac{\eta}{2} \operatorname{Tr} \left[ \boldsymbol{\Sigma}_\varepsilon \nabla^2 L_\delta \right]$$

$$= -\langle \nabla L + 20\delta |\boldsymbol{\theta}|^{18} \boldsymbol{\theta}, \ \nabla L + 20\delta |\boldsymbol{\theta}|^{18} \boldsymbol{\theta} \rangle + \frac{\eta}{2} \operatorname{Tr} \left[ \boldsymbol{\Sigma}_\varepsilon \nabla^2 L_\delta \right]$$

$$= -\langle \nabla L + 20\delta |\boldsymbol{\theta}|^{18} \boldsymbol{\theta}, \ \nabla L + 20\delta |\boldsymbol{\theta}|^{18} \boldsymbol{\theta} \rangle + \frac{1}{2} \operatorname{Tr} \left[ \boldsymbol{\Sigma}_\varepsilon \nabla^2 L_\delta \right] \quad (\text{as } \eta \leq 1).$$

By the polynomial bounds established above, the dominant term is $-400\delta^2 |\boldsymbol{\theta}|^{38}$, so that $\mathcal{A}V \to -\infty$ as $|\boldsymbol{\theta}| \to \infty$. Hence, there exists a constant $c > 0$ independent of $\eta$ such that $\mathcal{A}V \leq c$ for all $\boldsymbol{\theta} \in \mathbb{R}^M$. Since $V \geq 1$, it follows that $\mathcal{A}V \leq cV$. $\qquad \square$

The function $V$ introduced in the above lemma serves as a Lyapunov function for the SDE (10). As the proof illustrates, its existence crucially relies on the addition of the higher-order regularization term $\delta |\boldsymbol{\theta}|^{2s}$. The Lyapunov function plays a key role in the analysis of SDEs, and we will use it extensively to establish several results in the subsequent sections Khasminskii (2012); Yu et al. (2021). As a first application, it allows us to prove the existence and uniqueness of solutions to the SDE (10). We begin by introducing some necessary definitions.

**Definition 3** (Equivalent solutions). *Fix any $T > 0$. Two solutions $\boldsymbol{\Theta}_1(t)$ and $\boldsymbol{\Theta}_2(t)$ of some SDE are said to be equivalent in $[0, T]$ if*

$$\mathbb{P}\left(\boldsymbol{\Theta}_1(t) = \boldsymbol{\Theta}_2(t) \text{ for all } t \in [0, T]\right) = 1,$$

*which means that these two solutions are identical with probability 1 over the time interval $[0, T]$.*

**Definition 4** (Explosion time). *Let $\tau(\boldsymbol{\theta})$ denote the explosion time of the solution $\boldsymbol{\Theta}(t)$ with $\boldsymbol{\Theta}(0) = \boldsymbol{\theta}$ to Eq.(10), defined as $\lim_{R \to \infty} \tau_R(\boldsymbol{\theta})$, where*

$$\tau_R(\boldsymbol{\theta}) = \inf\{t \geq 0 : \boldsymbol{\Theta}(t) \notin B_R \text{ with } \boldsymbol{\Theta}(0) = \boldsymbol{\theta}\}$$

*is the first exit time of $\boldsymbol{\Theta}(t)$ from a ball of radius $R$.*

By applying Theorem 3.5 in Khasminskii (2012), together with the Lyapunov function constructed in Lemma 3, we can obtain the existence and uniqueness of solutions to the SDE (10) directly.

**Theorem 2** (rephrased from Theorem 3.5 in Khasminskii (2012)). *Under Assumptions 1 and 2, for the noisy regularized SDE* (10)*, we have the following conclusions:*

(1) *For every random variable $\boldsymbol{\Theta}(0)$ independent of the process $\boldsymbol{W}(t) - \boldsymbol{W}(0)$, there exists a solution $\boldsymbol{\Theta}(t)$ to Eq.(10) which is an almost surely continuous stochastic process and is unique up to equivalence.*

(2) *This solution is a Markov process whose Feller transition probability function $P(\boldsymbol{\theta}, t, A)$ is defined for $t > 0$ by $P(\boldsymbol{\theta}, t, A) = \mathbb{P}\{\boldsymbol{\Theta}(t) \in A \mid \boldsymbol{\Theta}(0) = \boldsymbol{\theta}\}$.*

(3) *This process satisfies the following inequality:*

$$\mathbb{E}[V(\boldsymbol{\Theta}_t)] \leq \mathbb{E}[V(\boldsymbol{\Theta}_0)]e^{ct}, \tag{11}$$

*where $V$ and $c$ are as defined in Lemma 3.*

(4) *The SDE* (10) *is complete, i.e.,*

$$\mathbb{P}(\tau(\boldsymbol{\theta}) = \infty) = 1, \quad \text{for all} \quad \boldsymbol{\theta} \in \mathbb{R}^M. \tag{12}$$

*Proof of Theorem 2.* Due to our modification of the diffusion term, the following conditions hold: for any $R > 0$, there exists a constant $B > 0$ such that for all $\boldsymbol{\theta}, \boldsymbol{\xi} \in B_R$,

$$|\boldsymbol{b}_\delta(\boldsymbol{\theta}) - \boldsymbol{b}_\delta(\boldsymbol{\xi})| + \sqrt{\eta}\|\boldsymbol{\sigma}_\varepsilon(\boldsymbol{\theta}) - \boldsymbol{\sigma}_\varepsilon(\boldsymbol{\xi})\|_F \leq B|\boldsymbol{\theta} - \boldsymbol{\xi}|$$

and

$$|\boldsymbol{b}_\delta(\boldsymbol{\theta})| + \sqrt{\eta}\|\boldsymbol{\sigma}_\varepsilon(\boldsymbol{\theta})\|_F \leq B(1 + |\boldsymbol{\theta}|).$$

Together with Lemma 3, these conditions allow us to apply Theorem 3.5 in Khasminskii (2012) directly to establish the result. $\qquad\square$

This theorem establishes the non-explosiveness of solutions, which is a property of fundamental importance. Ensuring non-explosiveness guarantees the long-term stability of the SDE, as the solution remains well-behaved for all time.

# D  SUPPLEMENTARY NOTES FOR PROPOSITION 1

In this section, we provide a detailed discussion of topics related to Proposition 1 and present its proof.

## D.1  NOTION OF WEAK CONVERGENCE

We begin by recalling the notion of weak convergence, which formalizes the convergence of probability laws through expectations of test functionals rather than pointwise trajectories.

Intuitively, a sequence of random variables (or stochastic processes) converges weakly if all bounded, sufficiently regular observables of the sequence converge in expectation to those of a limiting random variable (or process). Formally, let $\{X_n\}_{n \geq 1}$ be a sequence of random variables taking

values in a Polish space $(\mathcal{X}, \mathcal{B}(\mathcal{X}))$, and let $X$ be an $\mathcal{X}$-valued random variable. We say that $X_n$ converges weakly to $X$ (denoted $X_n \Rightarrow X$) if, for every bounded continuous function $\varphi : \mathcal{X} \to \mathbb{R}$,

$$\lim_{n \to \infty} \mathbb{E}\big[\varphi(X_n)\big] \;=\; \mathbb{E}\big[\varphi(X)\big].$$

Equivalently, the probability measures $\{\mathcal{L}(X_n)\}$ converge to $\mathcal{L}(X)$ in the topology of weak convergence.

In the context of numerical approximations to stochastic dynamics (e.g., discrete-time algorithms approximating SDEs), weak convergence of order $p > 0$ over a finite horizon $T$ typically means that, for all test functions $\varphi$ in a prescribed class (e.g., $\varphi \in G^\alpha$ for some $\alpha$), there exists $C_{\varphi,T} > 0$ such that

$$\big|\mathbb{E}[\varphi(X_h(T))] - \mathbb{E}[\varphi(X(T))]\big| \;\leq\; C_{\varphi,T}\, h^p,$$

where $X_h(T)$ denotes the numerical approximation at step size $h$, and $X(T)$ the exact solution at time $T$. This definition captures convergence at the level of distributions and observables, which is the relevant notion for many inference, optimization, and uncertainty quantification tasks.

## D.2 PRODUCT PROBABILITY SPACE

Establishing weak convergence between SGD and its SDE surrogate is central to justifying continuous-time approximations for discrete stochastic optimization: it guarantees that expectations of sufficiently regular observables computed along the SGD iterates converge to those of the limiting SDE, thereby validating distributional predictions (e.g., bias, variance, and risk) drawn from the continuous model for the discrete algorithm.

In our setting, however, the coefficients of SDE Eq.(10) are only locally (not globally) Lipschitz, so global arguments are unavailable and the analysis must be carried out locally. To this end, we place SGD and the SDE on a common product probability space, namely the product of the SGD sample space $(U^n)^\infty$ (encoding the i.i.d. mini-batch or gradient noise) and the SDE sample space, equipped with the product measure. The SGD sample space is $\Omega_{\mathrm{sgd}} := (U^n)^\infty$ with its cylinder $\sigma$-algebra $\mathcal{F}_{\mathrm{sgd}}$ and law $\mathbb{P}_{\mathrm{sgd}}$ (induced by the i.i.d. noise sequence). Let the SDE live on the Wiener space $(\Omega_{\mathrm{w}}, \mathcal{F}_{\mathrm{w}}, \mathbb{P}_{\mathrm{w}})$, where $\Omega_{\mathrm{w}} = C([0, \infty); \mathbb{R}^M)$ is the space of continuous paths, $\mathcal{F}_{\mathrm{w}}$ is the canonical $\sigma$-algebra, and $\mathbb{P}_{\mathrm{w}}$ is the Wiener measure under which the canonical process $W_t(\omega) = \omega(t)$ is an $M$-dimensional Brownian motion. We then define the product probability space

$$(\Omega, \mathcal{F}, \mathbb{P}) = \big(\Omega_{\mathrm{sgd}} \times \Omega_{\mathrm{w}},\ \mathcal{F}_{\mathrm{sgd}} \otimes \mathcal{F}_{\mathrm{w}},\ \mathbb{P}_{\mathrm{sgd}} \otimes \mathbb{P}_{\mathrm{w}}\big).$$

Under this construction, the two noise sources remain independent, while both processes can be viewed as random elements on the same probability space. This facilitates the definition of stopping times that confine the dynamics to regions where local Lipschitz and polynomial growth controls hold, enabling a rigorous, localized weak convergence analysis.

## D.3 PROOF OF PROPOSITION 1

Before proving Proposition 1, we present a key weak convergence result established in Mil'shtein (1986).

**Theorem 3** (Theorem 2 in Mil'shtein (1986)). *Consider the SDE*

$$d\boldsymbol{\Theta}_t = \boldsymbol{b}(\boldsymbol{\Theta}_t)dt + \boldsymbol{\sigma}(\boldsymbol{\Theta}_t)d\boldsymbol{W}(t).$$

*Assume that this SDE is well-defined on interval $[0, T]$. Define its discrete approximation with step size $\eta$ as*

$$\boldsymbol{\theta}(0) = \boldsymbol{\Theta}(0),\ \boldsymbol{\theta}(k+1) = \boldsymbol{\theta}(k) + A(k\eta, \boldsymbol{\theta}(k), \eta; \xi),\ \forall k = 0, 1, ...$$

*where $\xi$ is some (generally vector) r.v. possessing sufficiently high moments, and $A$ is a vector function of dimension $M$. Let the following conditions hold:*

*(i) The coefficients of the SDE satisfy the global Lipschitz condition and, together with all their partial derivatives up to order $2p + 2$ inclusive, belong to the class $G$.*

*(ii) The discretization is such that*

$$\mathbb{E}\left|\prod_{j=1}^{s}\Delta_{i_j} - \prod_{j=1}^{s}\overline{\Delta}_{i_j}\right| \le K_1(\boldsymbol{\theta})\eta^{p+1}, \quad s = 1,\dots,2p+1, \quad K_1(\boldsymbol{\theta}) \in G,$$

$$\mathbb{E}\prod_{j=1}^{2p+2}|\overline{\Delta}_{i_j}| \le K_2(\boldsymbol{\theta})\eta^{p+1}, \quad K_2(\boldsymbol{\theta}) \in G,$$

*where $\Delta = \boldsymbol{\Theta}(\eta) - \boldsymbol{\Theta}(0)$, $\overline{\Delta} = \boldsymbol{\theta}(1) - \boldsymbol{\theta}(0)$, and $1 \le i_j \le M$.*

*(iii) For sufficiently large $m$, $\mathbb{E}|\boldsymbol{\theta}(k)|^{2m}$ exist and are uniformly bounded in $\eta$ and $k = 0, 1, \dots, \lfloor\frac{T}{\eta}\rfloor$.*

*If $f$ and all its partial derivatives in $\boldsymbol{\theta}$ up to order $2p+2$ inclusive belong to the class $G$, then for all $0 < \eta < 1$ and $k = 0, 1, \dots, \lfloor\frac{T}{\eta}\rfloor$,*

$$|\mathbb{E}f(\boldsymbol{\Theta}(k\eta)) - \mathbb{E}f(\boldsymbol{\theta}(k))| \le K\eta^p, \tag{13}$$

*where $K$ is a constant. That is, the method achieves an order-p accuracy in the sense of weak approximations.*

Now we use the above weak convergence result to prove Proposition 1.

*Proof of Proposition 1.* We follow the classical weak convergence framework for SDE discretizations introduced in Theorem 3, localizing the analysis to $B_R$ and leveraging local Lipschitz continuity to control the error.

Theorem 3 asserts that if the one-step weak error of the numerical method is controlled at order $\eta^{p+1}$ in expectation, and certain regularity and moment conditions are met, then the cumulative weak error after multiple steps is of order $\eta^p$. Here, we focus on the case $p = 1$.

We henceforth carry out all considerations on the previously defined product probability space. Below, we verify the required conditions in our setting:

(i) **Local Lipschitz regularity:** On the ball $B_R$, the coefficients of Eq.(10) and their derivatives up to order $4$ are locally Lipschitz and bounded, as required.

(ii) **One-step weak error:** Let $\boldsymbol{\Theta}^{(\eta)}(\eta)$ be the solution of the SDE after one time step $\eta$, and $\boldsymbol{\theta}^{(\eta)}(1)$ the outcome of one step of the discrete algorithm, both starting from $\boldsymbol{\theta}$:

$$\boldsymbol{\Theta}^{(\eta)}(\eta) = \boldsymbol{\theta} + \int_0^\eta \boldsymbol{b}_\delta\left(\boldsymbol{\Theta}^{(\eta)}(s)\right)\mathrm{d}s + \int_0^\eta \sqrt{\varepsilon}\,\boldsymbol{\sigma}_\varepsilon\left(\boldsymbol{\Theta}^{(\eta)}(s)\right)\mathrm{d}\boldsymbol{W}(s),$$

$$\boldsymbol{\theta}^{(\eta)}(1) = \boldsymbol{\theta} - \frac{\eta}{n}\sum_{i=1}^n \nabla_{\boldsymbol{\theta}}\ell_\delta\left(\boldsymbol{\theta}, \boldsymbol{x}_i^{(k)}\right) + \sqrt{\varepsilon}\,\boldsymbol{g}(k).$$

Denote $\Delta = \boldsymbol{\Theta}^{(\eta)}(\eta) - \boldsymbol{\theta}$ and $\bar{\Delta} = \boldsymbol{\theta}^{(\eta)}(1) - \boldsymbol{\theta}$.

Under the event $k\eta \le \tau_R^{(\eta)}(\boldsymbol{\theta})$ and the event $k\eta \le \Gamma_R^{(\eta)}(\boldsymbol{\theta})$, since $\boldsymbol{b}_\delta$ and $\boldsymbol{\sigma}_\varepsilon$ are bounded in $B_R$, we have for each $i = 1,\dots,M$,

$$\mathbb{E}\left[|\Delta_i|\right] \le C\eta, \quad \mathbb{E}\left[|\bar{\Delta}_i|\right] \le C\eta, \tag{14}$$

for some constant $C > 0$.

For any $i = 1, \cdots, M$, we can estimate the error between the $i$-th entry of $\Delta$ and $\bar{\Delta}$ as follows:

$$
\begin{aligned}
\left| \mathbb{E} \left[ \Delta_i - \bar{\Delta}_i \right] \right| &= \left| \mathbb{E} \left[ \int_0^\eta \partial_i L_\delta \left( \boldsymbol{\Theta}^{(\eta)}(s) \right) \mathrm{d}s - \frac{\eta}{n} \sum_{i=1}^n \partial_i \ell_\delta \left( \boldsymbol{\theta}, \boldsymbol{x}_i^{(k)} \right) \right] \right| \\
&= \left| \int_{U^n} \left[ \int_0^\eta \partial_i L_\delta \left( \boldsymbol{\Theta}^{(\eta)}(s) \right) \mathrm{d}s - \frac{\eta}{n} \sum_{i=1}^n \partial_i \ell_\delta \left( \boldsymbol{\theta}, \boldsymbol{x}_i^{(k)} \right) \right] \mathrm{d}\lambda \right| \\
&= \left| \int_{U^n} \int_0^\eta \int_U \partial_i \ell_\delta \left( \boldsymbol{\Theta}^{(\eta)}(s), \boldsymbol{x} \right) \mathrm{d}\boldsymbol{x} \, \mathrm{d}s \mathrm{d}\lambda - \eta \int_U \partial_i \ell_\delta \left( \boldsymbol{\theta}, \boldsymbol{x} \right) \mathrm{d}\boldsymbol{x} \right| \\
&= \left| \int_{U^n} \int_0^\eta \left( \int_U \partial_i \ell_\delta \left( \boldsymbol{\Theta}^{(\eta)}(s), \boldsymbol{x} \right) - \partial_i \ell_\delta \left( \boldsymbol{\theta}, \boldsymbol{x} \right) \mathrm{d}\boldsymbol{x} \right) \mathrm{d}s \mathrm{d}\lambda \right| \\
&\leq C_R \int_{U^n} \int_0^\eta \left| \boldsymbol{\Theta}^{(\eta)}(s) - \boldsymbol{\theta} \right| \mathrm{d}s \mathrm{d}\lambda \\
&\qquad \left( \text{by the Lipschitz continuity of } \int_U \partial_i \ell_\delta \left( \boldsymbol{\Theta}^{(\eta)}(s), \boldsymbol{x} \right) - \partial_i \ell_\delta \left( \boldsymbol{\theta}, \boldsymbol{x} \right) \mathrm{d}\boldsymbol{x} \right) \\
&\leq C_R C \int_{U^n} \int_0^\eta \eta \, \mathrm{d}s \mathrm{d}\lambda \quad (\text{by Eq.(14)}) \\
&\leq C_R C \eta^2.
\end{aligned}
$$

As for $s = 2, 3$, it is easy to follow that

$$
\left| \mathbb{E} \left( \prod_{j=1}^s \Delta_{i_j} - \prod_{j=1}^s \bar{\Delta}_{i_j} \right) \right| \leq C \eta^s,
$$

where $1 \leq i_j \leq M$ for $j = 1, \cdots, s$.

Moreover, it is straightforward that

$$
\mathbb{E} \prod_{j=1}^4 \left| \bar{\Delta}_{i_j} \right| \leq C \eta^2,
$$

where $1 \leq i_j \leq M$ for $j = 1, \cdots, 4$.

So condition (ii) in Theorem 2 of Mil'shtein (1986) for $p = 1$ holds.

(iii) **Moment bounds:** Both the SDE and the SGD iteration remain in $B_R$ for all steps under consideration, so the relevant moments are uniformly bounded in $k$ and $\eta$.

With these conditions verified, Theorem 3 directly yields that if $h \in G^6$, then for all $0 < \eta < 1$ and all $k$, we have

$$
\left| \mathbb{E} \left[ \left( h \left( \boldsymbol{\Theta}^{(\eta)}(k\eta) \right) - h \left( \boldsymbol{\theta}^{(\eta)}(k) \right) \right) 1_{\left\{ k\eta \leq \tau_R^{(\eta)}(\theta) \right\}} 1_{\left\{ k\eta \leq \Gamma_R^{(\eta)}(\theta) \right\}} \right] \right| \leq C(R, T) \eta.
$$

for some constant $C(R, T) > 0$ independent of $\eta$. This completes the proof.

$\square$

# E    EXPLANATION AND VALIDATION OF ASSUMPTION 3

To substantiate our theoretical analysis, we provide additional clarification and empirical validation regarding Assumption 3 (uniform moment bounds). Specifically, we conducted numerical experiments to examine the plausibility of this assumption. In our experiment, we use a physics-informed neural network (PINN) to solve a two-dimensional Helmholtz equation. We performed a series of training runs using 40 different step sizes uniformly spaced on a logarithmic scale from $10^{-5}$ to

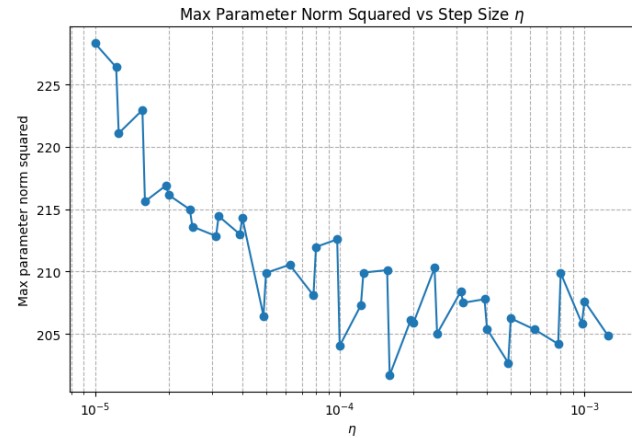

Figure 4: Maximal squared $\ell^2$-norm of parameters during training for various step sizes.

$10^{-3}$. For each configuration, training was conducted for $1/\eta$ steps, and we recorded the maximal squared $\ell^2$-norm of the parameters throughout training. The results, shown in Figure 4, reveal that as the step size decreases toward $10^{-5}$, the maximal parameter norms tend to increase. This suggests that verifying the uniform moment bound is primarily relevant for sufficiently small step sizes.

When the step size is sufficiently small, theoretical guarantees become available. For fixed $p$ and $\eta$, the recursion for the stochastic gradient descent (SGD) iterates can be expressed as

$$\mathbb{E}\|\theta_{k+1}\|^{2p} \leq \mathbb{E}\|\theta_k\|^{2p} + C\eta\big(1 + \mathbb{E}\|\theta_k\|^{2p+2(s-1)}\big).$$

As $\eta \to 0$, the discrete-time dynamics are controlled by the following ordinary differential equation (ODE):

$$\frac{du}{dt} = C\left(1 + u^{1+\frac{1}{p}(s-1)}\right),$$

for which standard theory ensures uniform boundedness on finite intervals $[0, T]$ for some $T > 0$. Therefore, uniform moment bounds can be guaranteed for sufficiently small step sizes.

Taken together, these theoretical and empirical findings support the practical validity of our uniform moment bound assumption.

## F    PROOF FOR SOME RESULTS IN SECTION 3

### F.1    PROOF OF PROPOSITION 2

*Proof of Proposition 2.* Note that

$$\mathbb{E}^{\boldsymbol{\theta}}[V(\boldsymbol{\Theta}^{(\eta)}(T))] = \mathbb{E}^{\boldsymbol{\theta}}\left[V(\boldsymbol{\Theta}^{(\eta)}(T))\mathbf{1}_{\tau_R^{(\eta)} \leq T} + V(\boldsymbol{\Theta}^{(\eta)}(T))\mathbf{1}_{\tau_R^{(\eta)} > T}\right]$$
$$\geq \mathbb{E}^{\boldsymbol{\theta}}[V(\boldsymbol{\Theta}^{(\eta)}(T))\mathbf{1}_{\tau_R^{(\eta)} \leq T}]$$
$$\geq \mathbb{P}\left(\tau_R^{(\eta)}(\boldsymbol{\theta}) \leq T\right)\inf_{|\boldsymbol{\xi}| \geq R} V(\boldsymbol{\xi}),$$

where $\mathbb{E}^{\boldsymbol{\theta}}$ means the expectation conditioned on $\boldsymbol{\Theta}^{(\eta)}(0) = \boldsymbol{\theta}$.

From Theorem 2, we have $\mathbb{E}^{\boldsymbol{\theta}}[V(\boldsymbol{\Theta}_t^{(\eta)})] \leq V(\boldsymbol{\theta})e^{ct}$ , so

$$\mathbb{P}\left(\tau_R^{(\eta)}(\boldsymbol{\theta}) \leq T\right) \leq \frac{e^{cT}V(\boldsymbol{\theta})}{\inf_{|\boldsymbol{\xi}| \geq R} V(\boldsymbol{\xi})} \to 0 \quad as \quad R \to \infty$$

because $V_R := \inf_{|\boldsymbol{\xi}| > R} V(\boldsymbol{\xi}) \to \infty$ as $R \to \infty$ (see Lemma 3). $\qquad\square$

### F.2 PROOF OF THEOREM 1

*Proof.* Fix $T > 0$, any bounded test function $h \in G^6$, $R > 0$, and initial value $\boldsymbol{\theta} \in \mathbb{R}^M$.

We decompose the total weak error into contributions inside and outside the ball $B_R$ up to time $T$:

$$\left| \mathbb{E}\, h\left( \boldsymbol{\Theta}^{(\eta)}(k\eta) \right) - \mathbb{E}\, h\left( \boldsymbol{\theta}^{(\eta)}(k) \right) \right|$$

$$\leq \left| \mathbb{E}\left[ \left( h\left( \boldsymbol{\Theta}^{(\eta)}(k\eta) \right) - h\left( \boldsymbol{\theta}^{(\eta)}(k) \right) \right) 1_{\left\{ k\eta \leq \tau_R^{(\eta)}(\theta) \right\}} 1_{\left\{ k\eta \leq \Gamma_R^{(\eta)}(\theta) \right\}} \right] \right|$$

$$+ \left| \mathbb{E}\left[ \left( h\left( \boldsymbol{\Theta}^{(\eta)}(k\eta) \right) - h\left( \boldsymbol{\theta}^{(\eta)}(k) \right) \right) \left( 1_{\left\{ k\eta \leq \tau_R^{(\eta)}(\theta) \right\}} + 1_{\left\{ k\eta \leq \Gamma_R^{(\eta)}(\theta) \right\}} \right) \right] \right|.$$

For the first term, we invoke Proposition 1, which shows that as long as the process does not exit $B_R$, the weak error is controlled:

$$\left| \mathbb{E}\left[ \left( h\left( \boldsymbol{\Theta}^{(\eta)}(k\eta) \right) - h\left( \boldsymbol{\theta}^{(\eta)}(k) \right) \right) 1_{\left\{ k\eta \leq \tau_R^{(\eta)}(\theta) \right\}} 1_{\left\{ k\eta \leq \Gamma_R^{(\eta)}(\theta) \right\}} \right] \right| \leq C(R,T)\eta.$$

For the second term, applying Proposition 2 and Proposition 3, we have uniform exit probability bounds:

$$\left| \mathbb{E}\left[ \left( h\left( \boldsymbol{\Theta}^{(\eta)}(k\eta) \right) - h\left( \boldsymbol{\theta}^{(\eta)}(k) \right) \right) \left( 1_{\left\{ k\eta \leq \tau_R^{(\eta)}(\theta) \right\}} + 1_{\left\{ k\eta \leq \Gamma_R^{(\eta)}(\theta) \right\}} \right) \right] \right|$$

$$\leq 2\|h\|_\infty \left[ \mathbb{P}\left( \tau_R^{(\eta)}(\theta) \leq T \right) + \mathbb{P}\left( \Gamma_R^{(\eta)}(\theta) \leq T \right) \right]$$

$$\leq 2\|h\|_\infty \left( \frac{e^{cT} V(\boldsymbol{\theta})}{\inf_{|\boldsymbol{\xi}| \geq R} V(\boldsymbol{\xi})} + \frac{C_T}{R^{2p}} \right).$$

Combining the above estimates yields

$$\left| \mathbb{E}\, h\left( \boldsymbol{\Theta}^{(\eta)}(k\eta) \right) - \mathbb{E}\, h\left( \boldsymbol{\theta}^{(\eta)}(k) \right) \right| \leq C(R,T)\eta + 2\|h\|_\infty \left( \frac{e^{cT} V(\boldsymbol{\theta})}{\inf_{|\boldsymbol{\xi}| \geq R} V(\boldsymbol{\xi})} + \frac{C_T}{R^{2p}} \right),$$

where $C(R,T)$ and $C_T$ are as in the referenced propositions/lemmas, and $c$ is from Proposition 2.

This completes the proof. $\qquad\square$

### F.3 PROOF OF PROPOSITION 4

*Proof.* Using Lemma 3, we can obtain that: there exists a $R > 0$ large enough such that

$$\mathcal{A}V \leq -1, \quad \text{for all} \quad \boldsymbol{\theta} \in B_R^c. \tag{15}$$

Define $\zeta_R(\theta) = \inf\{t \geq 0 : \boldsymbol{\Theta}(t) \in B_R \text{ with } \boldsymbol{\Theta}(0) = \boldsymbol{\theta}\}$.

Using the Theorem 3.9 in book Khasminskii (2012), we can know that

$$\mathbb{E}[\zeta_R(\boldsymbol{\theta})] \leq V(\boldsymbol{\theta}). \tag{16}$$

For every compact subset $K \subset \mathbb{R}^M$, $\sup_{\theta \in K} \mathbb{E}[\zeta_R(\boldsymbol{\theta})] \leq \sup_{\boldsymbol{\theta} \in K} V(\boldsymbol{\theta}) < \infty$.

Applying the Theorem 4.1 and Corollary 4.4 in Khasminskii (2012), we can conclude that SDE (10) has a unique invariant measure $\mu$. $\qquad\square$

## G FURTHER PROPERTIES OF THE SOLUTION TO THE NOISY REGULARIZED SDE

In this section, we examine properties of the noise-regularized SDE Eq.(10). While these results may not fully capture the discrete algorithm's dynamics, they provide valuable insights when the step size $\eta$ is sufficiently small.

### G.1 LOSS/MOMENT DECAY RATE

**Lemma 4** (Lyapunov stability bound). *For any fixed $c > 0$, there exists a positive constant $Q$ depending on $c$ such that*

$$\mathcal{A}V + cV \le Q, \quad \text{for all} \quad \boldsymbol{\theta} \in \mathbb{R}^M. \tag{17}$$

*Proof.* Referring to the proof of Lemma 3, we can easily derive the following conclusion:

$$\mathcal{A}V + cV \to -\infty, \quad \text{as} \quad |\boldsymbol{\theta}| \to \infty.$$

Thus, this theorem clearly holds because $V$ is smooth. $\square$

**Lemma 5** (Lyapunov decay rate). *The expected Lyapunov function decays at the following rate:*

$$\mathbb{E}^{\boldsymbol{\theta}} \left[ V \left( \boldsymbol{\Theta}(t) \right) \right] \le \frac{Q}{c} + \left( V(\boldsymbol{\theta}) - \frac{Q}{c} \right) e^{-ct}, \quad \text{for all } t > 0, \, \boldsymbol{\theta} \in \mathbb{R}^M, \tag{18}$$

*where $c$ and $Q$ are as defined in the previous lemma, and are independent of $\eta$. Here, $\mathbb{E}^{\boldsymbol{\theta}}$ denotes expectation conditioned on $\boldsymbol{\Theta}(0) = \boldsymbol{\theta}$.*

*Proof.* Using Dynkin's formula, we obtain that

$$\mathbb{E}^{\boldsymbol{\theta}}[V(\boldsymbol{\Theta}_t)] = V(\boldsymbol{\theta}) + \mathbb{E}^{\boldsymbol{\theta}} \left[ \int_0^t \mathcal{A}V(\boldsymbol{\Theta}_s) \, \mathrm{d}s \right].$$

Then we have

$$\frac{\mathrm{d}\mathbb{E}^{\boldsymbol{\theta}}[V(\boldsymbol{\Theta}_t)]}{\mathrm{d}t} = \mathbb{E}^{\boldsymbol{\theta}}[\mathcal{A}V(\boldsymbol{\Theta}_t)].$$

Recall that $\mathcal{A}V \le Q - cV$, so we have

$$\frac{\mathrm{d}\mathbb{E}^{\boldsymbol{\theta}}[V(\boldsymbol{\Theta}_t)]}{\mathrm{d}t} \le \mathbb{E}^{\boldsymbol{\theta}}[Q - cV(\boldsymbol{\Theta}_t)] = Q - c\mathbb{E}^{\boldsymbol{\theta}}[V(\boldsymbol{\Theta}_t)].$$

By Gronwall inequality, it can be concluded that

$$\mathbb{E}^{\boldsymbol{\theta}}[V \left( \boldsymbol{\Theta}(t) \right)] \le \frac{Q}{c} + \left( V(\boldsymbol{\theta}) - \frac{Q}{c} \right) e^{-ct}, \quad \forall \, t > 0, \quad \text{for all } \boldsymbol{\theta} \in \mathbb{R}^M.$$

$\square$

The above lemma provides an estimate for the exponential decay rate of the loss function $L_\delta = V - 1$ along trajectories of the SDE solution. Moreover, by an argument similar to the proof of Lemma 3, one can show that $|\cdot|^{2q}$ also serves as a Lyapunov function for Eq.(10) when $q \ge s$, and thus enjoys the same properties as described in the preceding two lemmas.

**Lemma 6** (Exponential decay of high-order moments). *Let $V_q = |\boldsymbol{\theta}|^{2q}$ for any integer $q \ge s$. Given $c > 0$, there exists a positive constant $Q$ depending on $c$ and $q$ such that*

$$\mathcal{A}V_q + cV_q \le Q, \quad \text{for all} \quad \boldsymbol{\theta} \in \mathbb{R}^M. \tag{19}$$

*Moreover,*

$$\mathbb{E} \left[ |\boldsymbol{\Theta}(t)|^{2q} \, \Big| \, \boldsymbol{\Theta}(0) = \boldsymbol{\theta} \right] \le \frac{Q}{c} + \left( |\boldsymbol{\theta}|^{2q} - \frac{Q}{c} \right) e^{-ct}, \quad \text{for all } t > 0, \, \boldsymbol{\theta} \in \mathbb{R}^M. \tag{20}$$

These results characterize that the expectations of the loss function and higher-order moment functions along the trajectory of the SDE solution decay approximately at an exponential rate, which ensures that, with a sufficiently small step size, SGD can essentially remain within a bounded region and achieve satisfactory learning performance.

## G.2 ERGODIC AND RECURRENCE PROPERTIES OF THE NOISE REGULARIZED SDE

With the correction to the diffusion term ($\mathbf{\Sigma}_\varepsilon = \mathbf{\Sigma} + \varepsilon \mathbf{I}$), the smallest eigenvalue of the diffusion matrix $\eta \mathbf{\Sigma}_\varepsilon$ is uniformly bounded away from zero. Moreover, Theorem 2 ensures that the solution to SDE (10) is a regular Markov process. By classical SDE theory, the solution thus enjoys several favorable properties that shed light on the dynamics of the discrete optimization process (9). We summarize these properties below and provide brief proofs.

**Proposition 6** (Positive recurrence). *The solution to the SDE* (10) *is a positive recurrent process; that is, for any bounded open set $A \subset \mathbb{R}^d$, the expected first return time*

$$\tau_A := \inf\{t > 0 : \mathbf{\Theta}(t) \in A\}$$

*is finite for all initial conditions $\mathbf{\Theta}(0) = \boldsymbol{\xi}$, i.e.,*

$$\mathbb{E}^{\boldsymbol{\xi}}[\tau_A] < \infty, \quad \text{for any } \boldsymbol{\xi} \in \mathbb{R}^M.$$

*Proof.* Using Lemma 3, we can obtain that: there exists a $R > 0$ large enough such that

$$\mathcal{A}V \le -1, \quad \text{for all} \quad \boldsymbol{\theta} \in B_R^c.$$

Referring to the Theorem 3.9 in book Khasminskii (2012), we can know that the solution to SDE (10) is a positive recurrent process according to the definition of positive recurrent process. $\square$

The following two results are immediate corollaries of the positive recurrence for Markov processes, as discussed in Khasminskii (2012), and we omit the proofs. We include them here to provide further insight into the dynamics of the solution to SDE (10).

**Corollary 1.** *Let $P(\boldsymbol{\theta}, t, \cdot)$ denote the transition probability of the solution to SDE* (10)*, i.e., $P(\boldsymbol{\theta}, t, A) = \mathbb{P}\{\mathbf{\Theta}(t) \in A \mid \mathbf{\Theta}(0) = \boldsymbol{\theta}\}$ for measurable sets $A$. Then the following properties hold:*

*(i) $\forall \boldsymbol{\theta} \in \mathbb{R}^M$, $\forall \epsilon > 0$, $\exists R > 0$, $t_0(\boldsymbol{\theta}) > 0$ such that $P(\boldsymbol{\theta}, t, B_R^c) < \epsilon$, $\forall t > t_0(\boldsymbol{\theta})$;*

*(ii) $\forall \alpha > 0$, $\exists \gamma_\alpha > 0$ s.t. $\forall \boldsymbol{\theta}, \boldsymbol{\theta}_0 \in B_R$, $\exists t_1(\boldsymbol{\theta}) > 0$ such that $P(\boldsymbol{\theta}, t, B_\alpha(\boldsymbol{\theta}_0)) > \gamma_\alpha$, $\forall t > t_1(\boldsymbol{\theta})$.*

Together, these results demonstrate the strong stability of the solution to SDE (10): the process remains confined within a sufficiently large region with high probability over time, and will enter any neighborhood within this region repeatedly, regardless of the initial condition. This reflects the strong recurrence and mixing properties of the system, ensuring stable long-term behavior. Moreover, invoking Theorem 2 of Hu et al. (2019), one can further gain precise insight into the dynamics of the SDE near non-degenerate local minimizers.

**Proposition 7** (rephrased from Theorem 2 in Hu et al. (2019)). *Suppose that $\boldsymbol{\theta}^*$ is a non-degenerate local minimizer of $L_\delta$. Then for any sufficiently small $r > 0$, there exists an open ball $B(\boldsymbol{\theta}^*, r)$ such that for any convex open set $D$ inside $B(\boldsymbol{\theta}^*, r)$ containing $\boldsymbol{\theta}^*$, there exists a constant $\bar{V}_D > 0$ depending only on $D$ such that the expected hitting time*

$$T^{\boldsymbol{\theta}} = \inf\{t \ge 0 : \mathbf{\Theta}(t) \in \partial D \text{ with } \mathbf{\Theta}(0) = \boldsymbol{\theta}\}$$

*satisfies*

$$\lim_{\eta \to 0^+} \eta \log[\mathbb{E}T^{\boldsymbol{\theta}}] = \bar{V}_D, \quad \text{for all} \quad \boldsymbol{\theta} \in D. \tag{21}$$

*Further, we have uniform control of the mean exit time: there exist $\varepsilon_1 \in (0, \varepsilon)$, $C_1, C_2 > 0$ and $\eta_0 > 0$ so that whenever $\eta \le \eta_0$,*

$$C_1 \le \inf_{\boldsymbol{\theta} \in B(\boldsymbol{\theta}^*, \varepsilon_1)} \eta \log[\mathbb{E}T^{\boldsymbol{\theta}}] \le \sup_{\boldsymbol{\theta} \in B(\boldsymbol{\theta}^*, \varepsilon_1)} \eta \log[\mathbb{E}T^{\boldsymbol{\theta}}] \le C_2. \tag{22}$$

*In particular, we define $N^{\boldsymbol{\theta}} = \frac{T^{\boldsymbol{\theta}}}{\eta}$. Then, there exist $C_3, C_4 > 0$ such that the expected steps needed to escape from a local minimizer satisfies*

$$C_3 \le \inf_{\boldsymbol{\theta} \in B(\boldsymbol{\theta}^*, \varepsilon_1)} \eta \log[\mathbb{E}N^{\boldsymbol{\theta}}] \le \sup_{\boldsymbol{\theta} \in B(\boldsymbol{\theta}^*, \varepsilon_1)} \eta \log[\mathbb{E}N^{\boldsymbol{\theta}}] \le C_4. \tag{23}$$

The preceding theorem shows that, on average, the system remains in the vicinity of a local minimizer for approximately $\exp(C\eta^{-1})$ steps before escaping. Proposition 7 extends the analysis of Theorem 2 in Hu et al. (2019): while Theorem 2 estimates the expected residence time of an SDE near a non-degenerate local minimizer, our result generalizes this by introducing additional parameters and more flexible conditions.

These properties of the SDE, such as positive recurrence and residence near local minima, offer valuable insight into the behavior of the noisy regularized SGD. They help explain, for small step sizes, how the algorithm is likely to explore the landscape and the typical timescale over which it escapes from local minima.

## H  WKB APPROXIMATION OF THE INVARIANT MEASURE

We have established that the SDE (10) admits a unique invariant ergodic measure $\mu$ with density $p$ relative to the Lebesgue measure. However, the exact form of $p$ is generally unknown due to the difficulty of solving the associated high-dimensional PDE. To address this, we apply the WKB approximation to derive an asymptotic representation of $p$ Miller Jr & Good Jr (1953); E et al. (2021).

Assume $p$ takes the form:

$$p(\boldsymbol{\theta}) = \exp\left(\frac{1}{\alpha}\sum_{n=0}^{\infty}\alpha^n S_n(\boldsymbol{\theta})\right). \tag{24}$$

Substituting Eq.(24) into Proposition 4, taking $\varepsilon = \frac{1}{n}$, and simplifying yields:

$$\frac{1}{2}\sum_{i,j=1}^{M}\left[\beta\partial^2_{\theta_i\theta_j}\hat{\Sigma}_{ij} + \beta\partial_{\theta_i}\hat{\Sigma}_{ij}\times\frac{1}{\alpha}\sum_{n=0}^{\infty}\alpha^n\partial_{\theta_j}S_n + \beta\partial_{\theta_j}\hat{\Sigma}_{ij}\times\frac{1}{\alpha}\sum_{n=0}^{\infty}\alpha^n\partial_{\theta_i}S_n\right.$$

$$\left.+\beta(\hat{\Sigma}_{ij}+\delta_{ij})\left(\frac{1}{\alpha^2}\sum_{n=0}^{\infty}\alpha^n\partial_{\theta_j}S_n\sum_{n=0}^{\infty}\alpha^n\partial_{\theta_i}S_n + \frac{1}{\alpha}\sum_{n=0}^{\infty}\alpha^n\partial^2_{\theta_i\theta_j}S_n\right)\right]$$

$$+\sum_{i=1}^{M}\left(\partial^2_{\theta_i\theta_i}L_\delta + \partial_{\theta_i}L_\delta\times\frac{1}{\alpha}\sum_{n=0}^{\infty}\alpha^n\partial_{\theta_i}S_n\right) = 0,$$

where $\beta = \frac{\eta}{n} = \eta\varepsilon$ represents the noise intensity, $\hat{\Sigma} = n\Sigma$, and $\delta_{ij}$ is the Kronecker delta. Setting $\alpha = \beta$ and collecting terms of the same order in $\beta$, we obtain a hierarchy of equations for $S_n$. The first three are:

$$\beta^{-1}:\left\langle\boldsymbol{b}_\delta - \frac{1}{2}(\hat{\boldsymbol{\Sigma}}+\boldsymbol{I})\nabla S_0, \nabla S_0\right\rangle = 0\ ;$$

$$\beta^0:\langle(\hat{\boldsymbol{\Sigma}}+\boldsymbol{I})\nabla S_0 - \boldsymbol{b}_\delta, \nabla S_1\rangle + \mathrm{Tr}\left(\nabla\hat{\boldsymbol{\Sigma}}\nabla S_0 + \frac{1}{2}(\hat{\boldsymbol{\Sigma}}+\boldsymbol{I})\nabla^2 S_0 + \nabla^2 L_\delta\right) = 0\ ;$$

$$\beta:\langle(\hat{\boldsymbol{\Sigma}}+\boldsymbol{I})\nabla S_0 - \boldsymbol{b}_\delta, \nabla S_2\rangle + \frac{1}{2}\langle\nabla S_1, (\hat{\boldsymbol{\Sigma}}+\boldsymbol{I})\nabla S_1\rangle$$

$$+\mathrm{Tr}\left(\nabla\hat{\boldsymbol{\Sigma}}\nabla S_1 + \frac{1}{2}(\hat{\boldsymbol{\Sigma}}+\boldsymbol{I})\nabla^2 S_1\right) + \frac{1}{2}\sum_{i,j=1}^{M}\partial^2_{\theta_i\theta_j}\hat{\Sigma}_{ij} = 0\ .$$

Analyzing these equations provides insight into the structure of $S_n$. In particular, for $S_0$, we have

$$\nabla S_0 = 2(\hat{\boldsymbol{\Sigma}}+\boldsymbol{I})^{-1}\boldsymbol{b}_\delta = -2(\hat{\boldsymbol{\Sigma}}+\boldsymbol{I})^{-1}\nabla L_\delta. \tag{25}$$

Thus,

$$S_0 = -2\int(\hat{\boldsymbol{\Sigma}}+\boldsymbol{I})^{-1}\,\mathrm{d}L_\delta. \tag{26}$$

Therefore, when $\beta \ll 1$, the WKB approximation yields the following asymptotic expression for $p$:

$$p \sim e^{\frac{1}{\beta}S_0(\boldsymbol{\theta})}, \tag{27}$$

where $\beta = \frac{\eta}{n}$ and $S_0$ is determined by Eq.(25).

**Remark 8.** *Under the isotropic assumption (i.e., $\Sigma = \beta I$), the WKB approximation can yield an explicit expression for the invariant measure $p$, which is consistent with the results reported in Dai & Zhu (2020).*

# I  INTRODUCTION TO LAPLACE METHOD

## I.1  PROOF OF LEMMA 2

*Proof.* Since $\nabla S_0 = 0$ if and only if $\nabla L_\delta = 0$, it suffices to show that $L_\delta$ has finitely many critical points. Under Assumption 1, each component of $\nabla L_\delta$ is analytic, so its zeros are isolated. Together with

$$|\nabla L_\delta(\boldsymbol{\theta})| \to \infty \quad \text{as} \quad |\boldsymbol{\theta}| \to \infty,$$

we conclude that $L_\delta$ has only finitely many critical points. $\qquad\square$

## I.2  LAPLACE METHOD

The Laplace method provides asymptotic approximations for integrals whose dominant contributions come from neighborhoods of maximizers (or minimizers) of a smooth phase function. It is especially effective when a small parameter $\beta > 0$ appears in the exponential, creating sharp concentration.

**Basic one-point formula.** Let $S \in C^2(\mathbb{R}^M)$ and $L \in C^0(\mathbb{R}^M)$, and assume: (i) $S$ attains its global maximum at an interior point $\theta^\star \in \mathbb{R}^M$, (ii) $\nabla S(\theta^\star) = 0$ and $H^\star := \nabla^2 S(\theta^\star)$ is negative definite, (iii) there exists $c > 0$ such that $S(\theta) \le S(\theta^\star) - c\|\theta - \theta^\star\|^2$ for $\|\theta - \theta^\star\|$ large (or more generally $S(\theta) < S(\theta^\star)$ away from $\theta^\star$ with sufficient decay to justify dominated convergence). Then, as $\beta \to 0$,

$$\int_{\mathbb{R}^M} L(\theta)\, e^{S(\theta)/\beta}\, \mathrm{d}\theta \;\sim\; (2\pi\beta)^{\frac{M}{2}} \frac{L(\theta^\star)\, e^{S(\theta^\star)/\beta}}{\left|-H^\star\right|^{1/2}},$$

where $|\cdot|$ denotes the determinant. Equivalently, since $H^\star$ is negative definite,

$$\left|-H^\star\right|^{1/2} = \left|\nabla^2 S(\theta^\star)\right|^{1/2}.$$

**Derivation (sketch).** Write a second-order Taylor expansion of $S$ at $\theta^\star$:

$$S(\theta) = S(\theta^\star) + \tfrac{1}{2}(\theta - \theta^\star)^\top H^\star(\theta - \theta^\star) + r(\theta), \qquad \frac{r(\theta)}{\|\theta - \theta^\star\|^2} \to 0.$$

Similarly, $L(\theta) = L(\theta^\star) + o(1)$ near $\theta^\star$. Split the integral into a small ball around $\theta^\star$ and its complement. The complement contributes a lower order term due to $S(\theta) < S(\theta^\star)$. Inside the ball, change variables $z = \beta^{-1/2}(\theta - \theta^\star)$ and neglect $r(\theta)$ at leading order, yielding a Gaussian integral:

$$\int_{\mathbb{R}^M} \exp\left(\tfrac{1}{2\beta}(\theta - \theta^\star)^\top H^\star(\theta - \theta^\star)\right) \mathrm{d}\theta = (2\pi\beta)^{\frac{M}{2}} |-H^\star|^{-1/2}.$$

Multiplying by $L(\theta^\star)e^{S(\theta^\star)/\beta}$ gives the formula.

**Multiple maximizers.** If $S$ attains its global maximum at finitely many nondegenerate points $\{\theta^{(j)}\}_{j=1}^J$ with negative-definite Hessians $H^{(j)} = \nabla^2 S(\theta^{(j)})$, then

$$\int_{\mathbb{R}^M} L(\theta)\, e^{S(\theta)/\beta}\, \mathrm{d}\theta \;\sim\; (2\pi\beta)^{\frac{M}{2}} \sum_{j=1}^J \frac{L(\theta^{(j)})\, e^{S(\theta^{(j)})/\beta}}{\left|-H^{(j)}\right|^{1/2}}, \qquad \beta \to 0.$$

**Application to our setting.** In our context, $p(\theta) \propto e^{S_0(\theta)/\beta}$ with $\beta = \eta/n$, $S_0 \in C^2$, and the set of maximizers of $S_0$ is finite and nondegenerate (see Lemma 2). Taking $L(\theta)$ as the loss, the Laplace method yields

$$\int_{\mathbb{R}^M} L(\theta)\, p(\theta)\, \mathrm{d}\theta \;\sim\; (2\pi\beta)^{\frac{M}{2}} \sum_j \frac{L(\theta^{(j)})\, e^{S_0(\theta^{(j)})/\beta}}{\left|\nabla^2 S_0(\theta^{(j)})\right|^{1/2}}, \qquad \beta \to 0.$$

With a unique nondegenerate maximizer $\theta^\star$,

$$\int e^{S_0(\theta)/\beta} d\theta \sim (2\pi\beta)^{M/2} \frac{e^{S_0(\theta^\star)/\beta}}{\left|\nabla^2 S_0(\theta^\star)\right|^{1/2}},$$

$$\int L(\theta)\, e^{S_0(\theta)/\beta} d\theta \sim (2\pi\beta)^{M/2} \frac{L(\theta^\star)\, e^{S_0(\theta^\star)/\beta}}{\left|\nabla^2 S_0(\theta^\star)\right|^{1/2}}.$$

Taking the ratio gives

$$\mathbb{E}_p[L(\theta)] \;=\; \frac{\int L(\theta)\, e^{S_0(\theta)/\beta} d\theta}{\int e^{S_0(\theta)/\beta} d\theta} \;\sim\; L(\theta^\star), \qquad \beta \to 0.$$

## J EVALUATION OF NOISY REGULARIZED SGD

To offer practical insights for readers who may be unfamiliar with the noisy regularized SGD algorithm, we present a series of experiments on several representative problems. In particular, we apply the algorithm to the Helmholtz equation, the Fisher–KPP equation, and the Allen–Cahn equation. These experiments are designed to illustrate the effectiveness and practical performance of the proposed approach across a range of partial differential equations. In addition, all experiments in the paper were conducted on a desktop computer equipped with a single 4060Ti GPU.

**Experiment 1: Helmholtz equation**

We first solve the following Helmholtz equation using the PINN framework:

$$\Delta u(x,y) + u(x,y) = f(x,y), \qquad (x,y) \in [-1,1] \times [-1,1],$$

where the exact solution is prescribed as $u^*(x,y) = \sin(\pi x)\sin(\pi y)$, which determines the source term $f(x,y)$ and the corresponding Dirichlet boundary conditions.

The neural network architecture employed is a fully connected network with three hidden layers, each containing 100 neurons. The training dataset consists of 10,000 collocation points in the interior and 400 points on the boundary. We use the noisy regularized SGD algorithm for training, where the regularization term is given by $\delta|\boldsymbol{\theta}|^{20}$ with $\delta = 10^{-30}$, and Gaussian noise $\sqrt{\varepsilon}\boldsymbol{g}(k)$ with $\varepsilon = 10^{-8}$. The batch size for interior points is set to 256, and for boundary points to 32. The step size is set to 0.005, and the model is trained for 1,000,000 iterations. After training, the predicted solution achieves a relative $L^2$ error of 0.00531 with respect to the exact solution. Figure 5 displays the predicted solution, the exact solution, and the pointwise error.

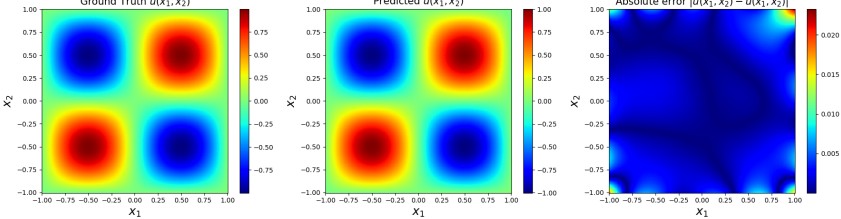

Figure 5: Predicted solution (left), exact solution (middle), and pointwise error (right) for the Helmholtz equation.

**Experiment 2: Fisher–KPP equation**

We present experiments on the classical two-dimensional Fisher–KPP equation, a widely studied reaction-diffusion model. The equation is given by

$$u_t = \Delta u + u(1-u) + S(x,y,t),$$

where $(x, y) \in [-1, 1]^2$ and $t \in [0, 1]$. The exact solution is selected as $u(x, y, t) = e^{-(x^2+y^2+t)}$, from which the source term $S(x, y, t)$ as well as the initial and boundary conditions can be directly determined.

Within the PINN framework, we employ a fully connected neural network with three hidden layers, each containing 50 neurons. The training set comprises 20,000 interior points and 500 boundary points, with batch sizes of 256 and 32 for interior and boundary points, respectively. Training is conducted using the noisy regularized SGD algorithm, where the regularization term is $\delta|\boldsymbol{\theta}|^{20}$ with $\delta = 10^{-30}$, and the Gaussian noise term $\sqrt{\varepsilon}\boldsymbol{g}(k)$ is incorporated with $\varepsilon = 10^{-10}$. The learning rate is set to 0.01, and the network is trained for 500,000 iterations. After training, the predicted solution attains a relative $L^2$ error of 0.00617 with respect to the exact solution. Figure 6 shows the predicted solution, the exact solution, and the corresponding pointwise error at three representative time points. Additionally, Figure 7 presents the training loss, which decreases rapidly and steadily

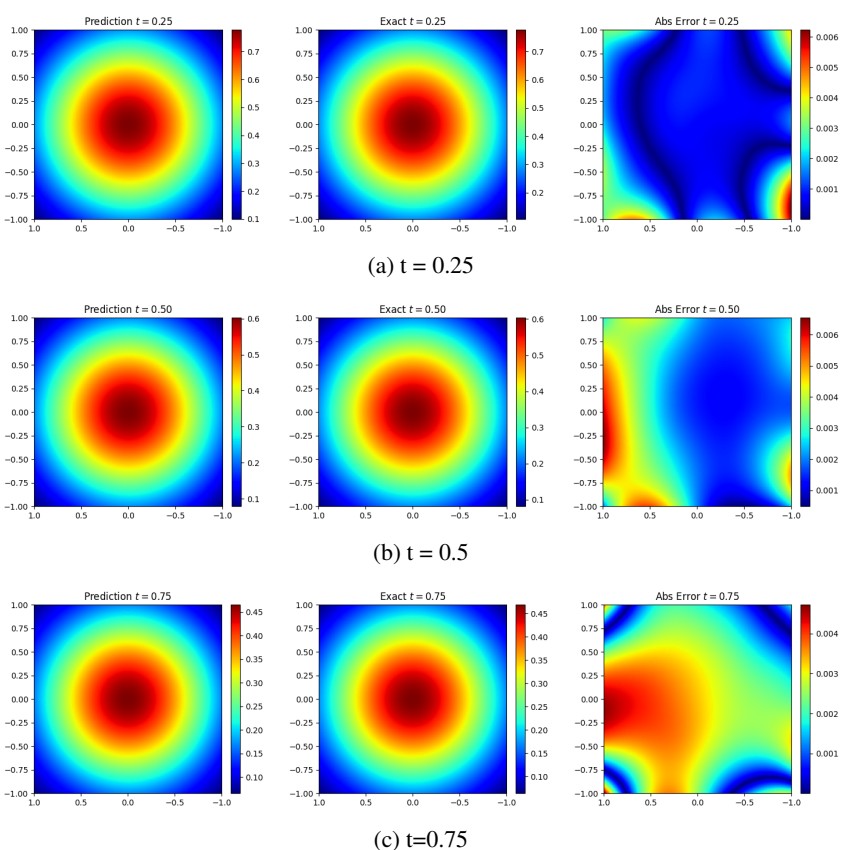

(a) t = 0.25

(b) t = 0.5

(c) t=0.75

Figure 6: Visualization of the predicted solution, the exact solution, and the pointwise error at three different time points for the Fisher–KPP equation.

throughout the optimization process.

**Experiment 3: Allen–Cahn equation**

We consider the two-dimensional Allen–Cahn equation,

$$u_t = \epsilon^2 \Delta u - (u^3 - u) + S(x, y, t),$$

on $(x, y) \in [-1, 1] \times [-1, 1]$, $t \in [0, 1]$, with $\epsilon = 0.1$. The exact solution we set is

$$u(x, y, t) = [\sin(\pi x)\cos(\pi y) + 0.1\sin(10\pi x)\cos(10\pi y)]e^{-t},$$

from which $S(x, y, t)$, initial, and boundary conditions are determined.

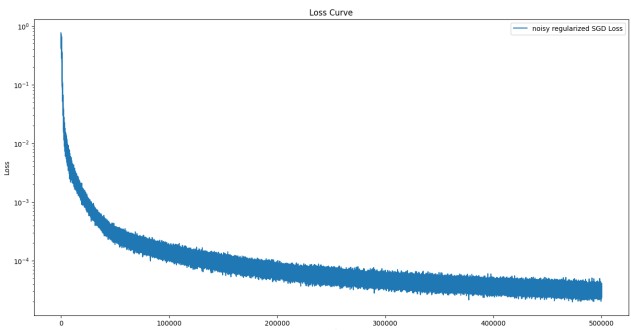

Figure 7: Loss evolution during training for the Fisher–KPP equation.

Within the PINN framework, we employ a fully connected neural network with three hidden layers, each comprising 100 neurons. The training dataset consists of 20,000 interior points and 500 boundary points, with batch sizes of 256 and 32 for interior and boundary points, respectively. Training is performed using the noisy regularized SGD algorithm, where the regularization term is $\delta|\boldsymbol{\theta}|^{20}$ with $\delta = 10^{-30}$, and the Gaussian noise term $\sqrt{\varepsilon}\boldsymbol{g}(k)$ is added with $\varepsilon = 10^{-10}$. The step size is set to $0.005$, and the network is trained for $1,000,000$ iterations. After training, the predicted solution achieves a relative $L^2$ error of $0.11234$ compared to the exact solution. Figure 8 illustrates the predicted solution, the exact solution, and the pointwise error at three different time points.

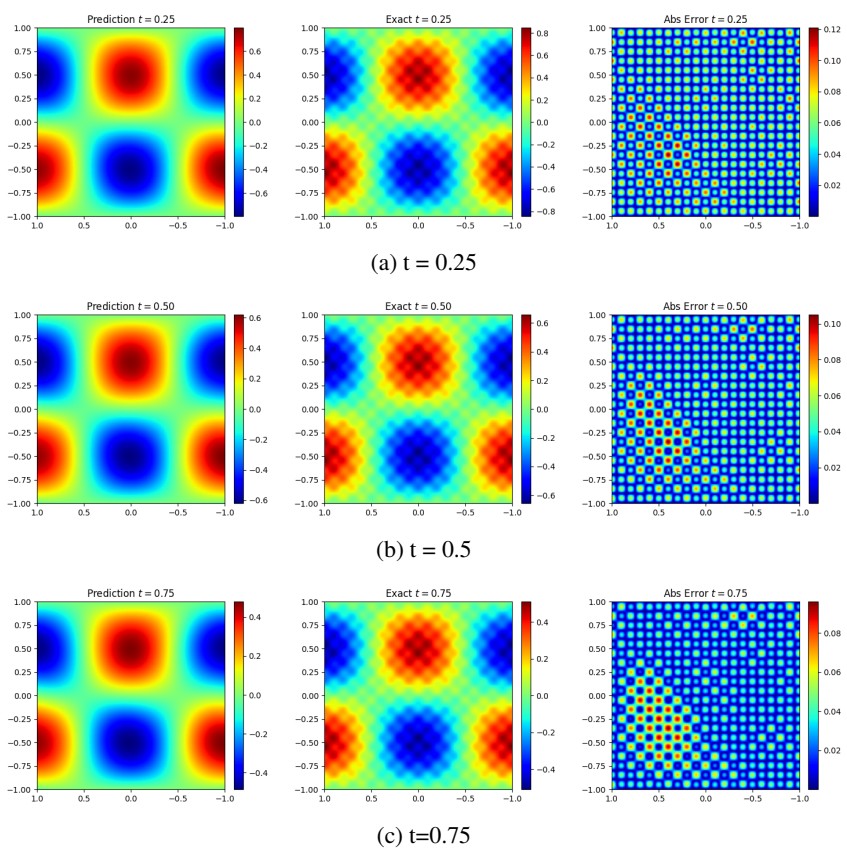

(a) t = 0.25

(b) t = 0.5

(c) t=0.75

Figure 8: Visualization of the predicted solution, the exact solution, and the pointwise error at three different time points for the Allen–Cahn equation.

We acknowledge that the achieved solution accuracy for this problem is relatively limited. One reason is the computational constraints, which prevent us from performing more extensive training; in some existing works, for example, over five million iterations are used to solve the Allen–Cahn equation. In addition, the inherent multiscale nature of the Allen–Cahn equation makes it particularly challenging to solve with standard approaches and often requires specialized techniques to achieve higher accuracy.

## K    DETAILED DESCRIPTION OF THE EXPERIMENTS: IMPACT OF STOCHASTICITY ON SGD PERFORMANCE

To facilitate our investigation into the role of stochasticity, we consider a second-order ordinary differential equation problem,

$$u''(x) = f(x), \quad x \in [-1, 1],$$

where the exact solution is prescribed as $u(x) = \tanh(2x + 1)$. The corresponding source term $f(x)$ and boundary conditions are derived from this choice. Importantly, this solution can be exactly represented by our chosen neural network architecture, allowing us to explicitly characterize all global minimizers in the parameter space.

Within the PINN framework, we employ a fully connected two-layer neural network of width 10, to approximate the solution:

$$u(x; \boldsymbol{\theta}) = \sum_{k=1}^{10} a_k \tanh\left(w_k x + b_k\right).$$

The training data consists of $1,000$ points uniformly sampled from the interval $[-1, 1]$.

Our goal is to experimentally investigate the impact of stochasticity in SGD by comparing the performance of SGD and deterministic GD under identical settings. Specifically, we focus on two distinct regimes: (1) near a global minimizer with small sharpness, and (2) near a global minimizer with large sharpness. Our results reveal that the relative advantages of SGD and GD vary significantly between these two regimes.

### K.1    REGIME 1: NEAR A GLOBAL MINIMIZER WITH SMALL SHARPNESS

In this experiment, we study the performance of SGD and GD in the vicinity of a global minimizer $\boldsymbol{\theta}^*$, where the parameter vector is given by $a_0 = 1$, $w_0 = 2$, $b_0 = 1$ for the first neuron and all other parameters are set to zero. The sharpness at this point is $31.14363$, yielding a theoretical critical learning rate for gradient descent of $\eta^{**} = 0.06422$.

#### K.1.1    EXPERIMENT 1: STABILITY DOMAINS OF LEARNING RATES FOR SGD AND GD

To investigate the stability properties, we conduct $50$ independent trials. In each run, the parameters are initialized randomly in a neighborhood of $\boldsymbol{\theta}^*$. Specifically, the first neuron is set as previously described, while the remaining parameters are initialized with Gaussian noise of mean $0$ and standard deviation $10^{-8}$. This initialization ensures that the parameters are very close to $\boldsymbol{\theta}^*$, and consequently, the sharpness closely matches that at $\boldsymbol{\theta}^*$. For each initialization, we consider a grid of $50$ learning rates, uniformly spaced from $0.001$ to $1.5\,\eta^*$. Both SGD (with batch size 32) and full-batch GD are run for $300$ steps at each learning rate. After training, we record the loss at the $300$-th step for both algorithms across all learning rates. For greater clarity, we present the pseudocode Algorithm 1.

The results consistently show that GD remains stable for all learning rates tested, never diverging in any of the $50$ runs. In contrast, SGD becomes unstable and exhibits explosion when the learning rate exceeds a certain threshold, a phenomenon observed in every trial. Here, "explosion" refers to a sharp increase in the loss at the $300$-th iteration compared to the loss observed at the $300$-th iteration with smaller learning rates. The Representative result from one typical run is shown in Figure 9, clearly illustrating that the stability regime for SGD is substantially smaller than that of GD, which is consistent with the result of Wu et al. (2018). Consequently, careful tuning of the learning rate is necessary for SGD in this setting.

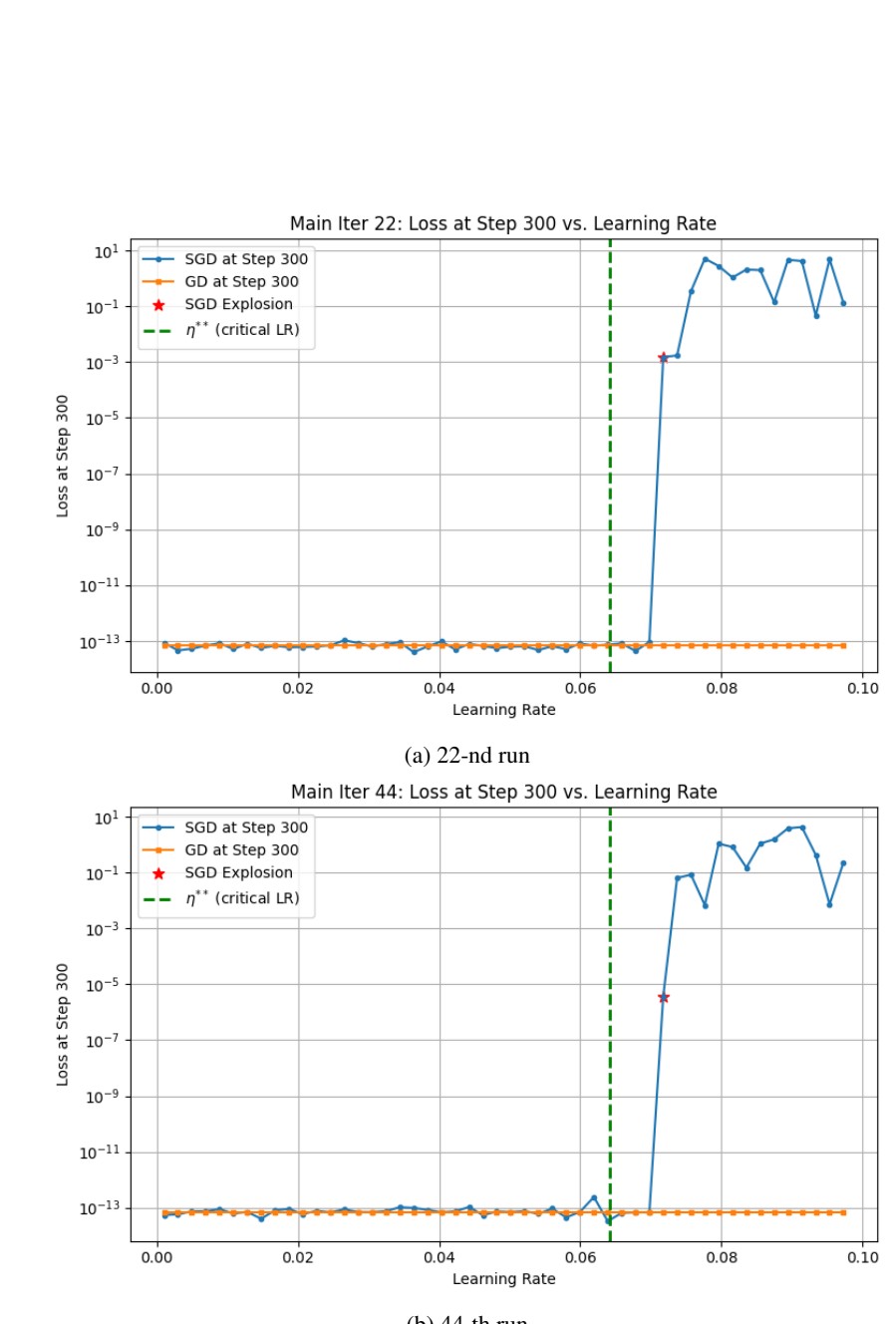

(a) 22-nd run

(b) 44-th run

Figure 9: Two typical runs comparing the stability domains of SGD and GD near $\theta^*$.

---

**Algorithm 1** Experiment 1: Stability domains of learning rates for SGD and GD

---

1: **Input:**
   Number of trials $N_{\text{trial}} = 50$;
   Learning rate grid $\{\eta_j\}_{j=1}^{50}$ uniformly spaced in $[0.001, 1.5\,\eta^*]$;
   Number of training steps $T = 300$, Batch size for SGD: 32.
2: **for** $i = 1$ **to** $N_{\text{trial}}$ **do**
3:   Initialize parameters $\boldsymbol{\theta}^{(i)}$ in a neighborhood of $\boldsymbol{\theta}^*$:
     • Set the first neuron as described in the main text
     • Initialize remaining parameters with Gaussian noise $\mathcal{N}(0, 10^{-16})$
4:   **for** $j = 1$ **to** $50$ **do**
5:     Set learning rate $\eta = \eta_j$
6:     $\boldsymbol{\theta}_{\text{SGD}}^{(i)} \leftarrow \boldsymbol{\theta}^{(i)}$
7:     **for** $t = 1$ **to** $T$ **do**
8:       Sample a mini-batch of size 32
9:       Update $\boldsymbol{\theta}_{\text{SGD}}^{(i)}$ by one step of SGD with step size $\eta$
10:    **end for**
11:    Record $\text{Loss}_{\text{SGD}}[i, j]$ as the loss at step $T$
12:    $\boldsymbol{\theta}_{\text{GD}}^{(i)} \leftarrow \boldsymbol{\theta}^{(i)}$
13:    **for** $t = 1$ **to** $T$ **do**
14:      Compute the full gradient over the dataset
15:      Update $\boldsymbol{\theta}_{\text{GD}}^{(i)}$ by one step of GD with step size $\eta$
16:    **end for**
17:    Record $\text{Loss}_{\text{GD}}[i, j]$ as the loss at step $T$
18:  **end for**
19: **end for**
20: **Output:** Loss arrays $\text{Loss}_{\text{SGD}}$ and $\text{Loss}_{\text{GD}}$ for all runs and learning rates

---

To further investigate the stability in practice, we record the critical learning rate at which SGD first diverges in each of the 50 independent runs. The distribution of these critical step sizes is shown in Figure 10.

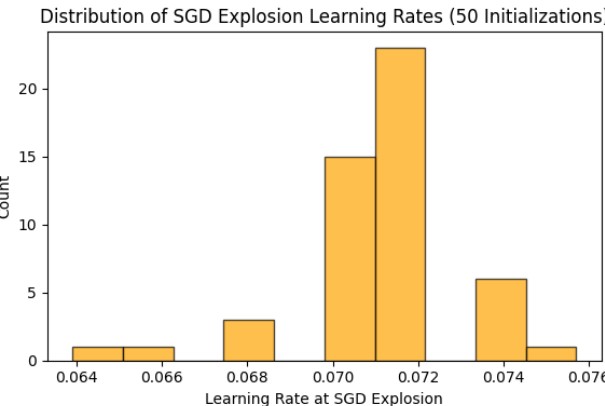

Figure 10: Distribution of the critical learning rates for SGD over 50 independent runs.

**Remark 9.** *Although the theoretical stable step size for GD is $\eta^{**}$, in practice, training often remains stable with a slightly larger step size (e.g., $1.5\eta^{**}$), because the theoretical bound is conservative and actual parameter updates rarely reach the worst-case curvature assumed in the analysis.*

### K.1.2 EXPERIMENT 2: EFFECT OF STOCHASTICITY WITHIN THE STABLE LEARNING RATE REGIME

We further investigate the effect of stochasticity within the stable domain of learning rates. In this experiment, we consider five increasing learning rates, 0.001, 0.002, 0.003, 0.004, and 0.005, all of which are smaller than $\eta^{**}$. For each learning rate, we conduct 50 independent runs. In each run, the parameters are initialized by adding noise to $\boldsymbol{\theta}^*$, where the covariance matrix of the noise is given by $\eta \boldsymbol{\Sigma}(\boldsymbol{\theta}^*)$. SGD with batch size 32 is then run for 3000 iterations. For greater clarity, we present the pseudocode Algorithm 2.

---

**Algorithm 2** Experiment 2: Effect of stochasticity within the stable learning rate regime

---

1: **Input:**
    Learning rates $\{\eta_j\}_{j=1}^5 = \{0.001,\ 0.002,\ 0.003,\ 0.004,\ 0.005\}$;
    Number of trials $N_{\text{trial}} = 50$; Number of training steps $T = 3000$;
    Batch size for SGD: 32; Reference parameter $\boldsymbol{\theta}^*$; Covariance function $\boldsymbol{\Sigma}(\boldsymbol{\theta}^*)$.
2: **for** $j = 1$ **to** 5 **do**
3:     Set learning rate $\eta = \eta_j$
4:     **for** $i = 1$ **to** $N_{\text{trial}}$ **do**
5:         Sample initial parameters $\boldsymbol{\theta}^{(i,j)}$ from $\mathcal{N}\left(\boldsymbol{\theta}^*,\ \eta\,\boldsymbol{\Sigma}(\boldsymbol{\theta}^*)\right)$
6:         $\boldsymbol{\theta}_{\text{SGD}} \leftarrow \boldsymbol{\theta}^{(i,j)}$
7:         **for** $t = 1$ **to** $T$ **do**
8:             Sample a mini-batch of size 32
9:             Update $\boldsymbol{\theta}_{\text{SGD}}$ by one step of SGD with step size $\eta$
10:         **end for**
11:         Record $\text{Output}_{\text{SGD}}[i, j]$ as the prediction on test points at step $T$
12:     **end for**
13: **end for**
14: **Output:** Output array $\text{Output}_{\text{SGD}}$ for all runs and learning rates

---

After training, for each of the 50 runs and each learning rate, we record the output of the learned function on 10,000 test points uniformly sampled from the interval $[-1, 1]$. For each learning rate, we take the mean of the outputs from the 50 runs as the solution learned by SGD at that step size. We then compute the variance and standard deviation of the 50 outputs with respect to this mean function. The results for the five learning rates are summarized as follows.

First, Table 1 in the main text reports the $L^2$ error of the mean function with respect to the ground truth at each learning rate.

Next, the variance and standard deviation curves of the 50 outputs relative to the mean function are depicted in Figure 11. Additionally, we plot the mean function along with the standard deviation band, as well as the ground truth function, in Figure 12. To further highlight the impact of stochasticity, we performed 50 runs of gradient descent (GD) with a learning rate of 0.005 under the same experimental setup. The relative $L^2$ error of the mean output function across these GD runs is $7.451 \times 10^{-6}$, which is an order of magnitude smaller than that obtained by SGD at the same learning rate. This striking contrast underscores how the inherent randomness in SGD significantly impacts both the learned solutions and the magnitude of their fluctuations.

These results provide a comprehensive perspective on both the learned solutions and the magnitude of their fluctuations under varying levels of stochasticity. As the learning rate increases, the impact of stochasticity becomes more apparent, since the covariance matrix $\eta \boldsymbol{\Sigma}$ is directly scaled by $\eta$ itself. This amplification of randomness is reflected in the output functions: at learning rates of 0.004 and 0.005, the variance among the 50 runs increases by one or more orders of magnitude. This substantial increase in variance leads to a notable decline in the accuracy of the mean solution at these higher learning rates. Overall, these results highlight the crucial influence of stochastic fluctuations on both the variability and reliability of the solutions learned by SGD within the stable learning rate regime.

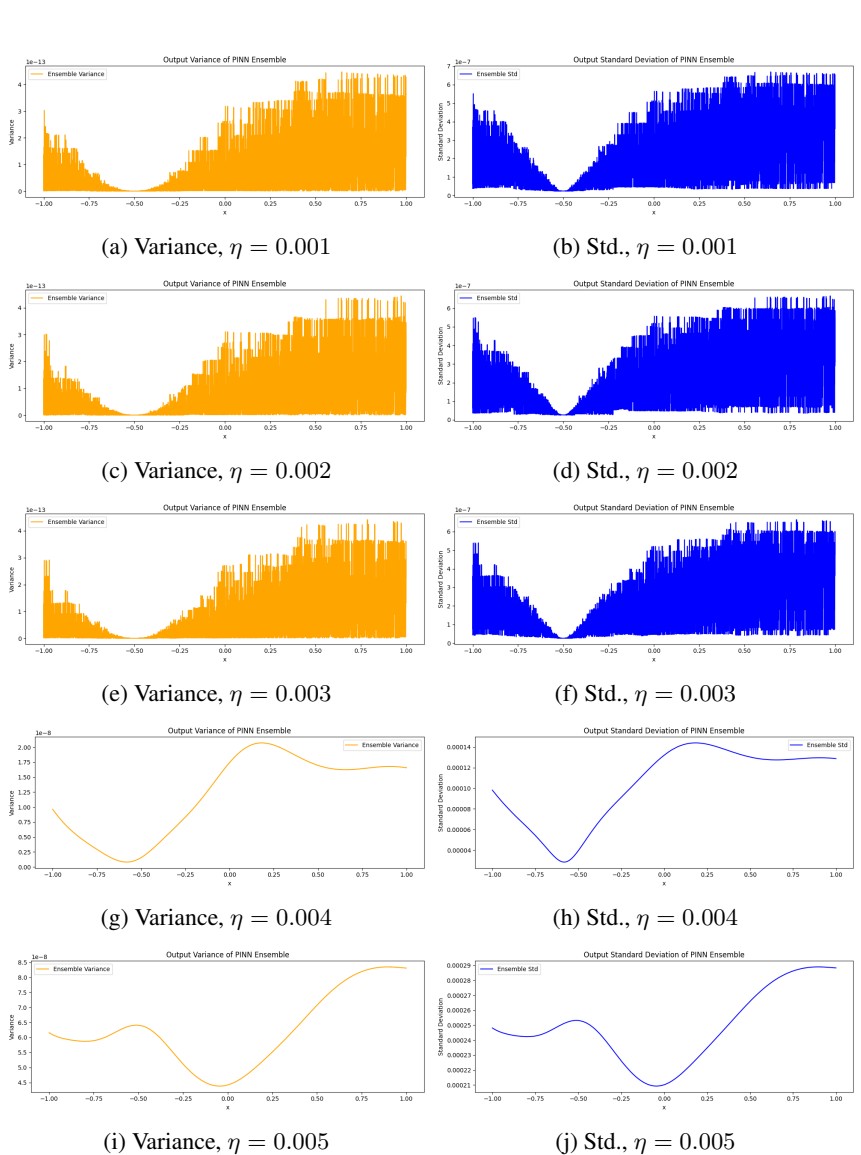

Figure 11: Variance and standard deviation curves of the 50 SGD outputs relative to the mean function, for each learning rate.

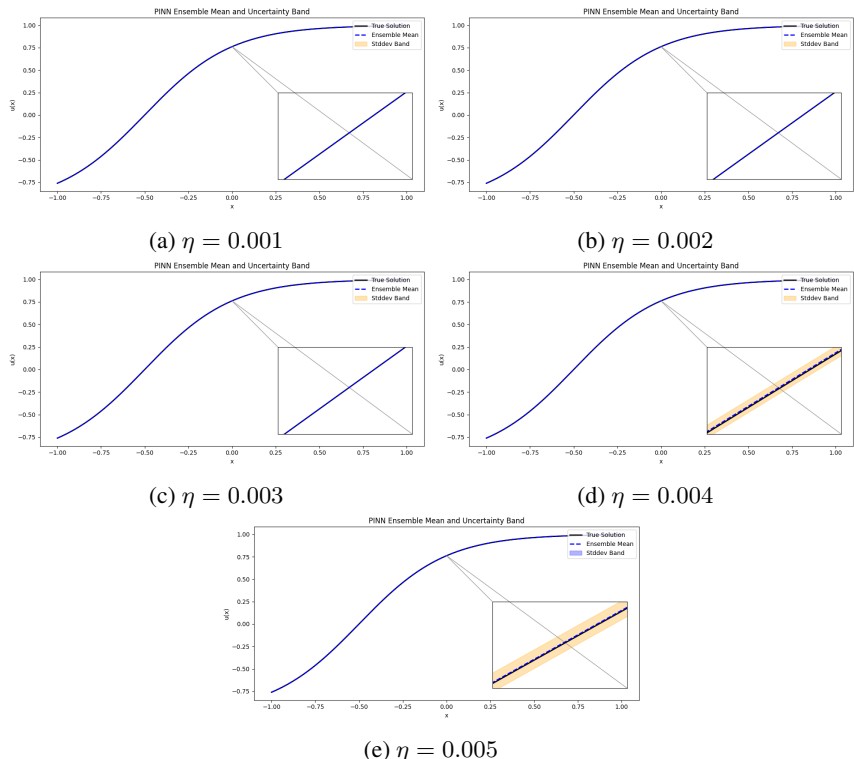

Figure 12: Mean function (solid line), standard deviation band (shaded area), and ground truth (dashed line) for each learning rate.

### K.2 REGIME 2: NEAR A GLOBAL MINIMIZER WITH LARGE SHARPNESS

Specifically, we select a global minimizer $\theta^{**}$ with large sharpness, constructed as in Table 2. At

Table 2: Construction of the global minimizer $\theta^{**}$

| $k$ | $a_k$ | $w_k$ | $b_k$ |
|---|---|---|---|
| 0 | 51.0 | 2.0 | 1.0 |
| 1, 2 | −25.0 | 2.0 | 1.0 |
| $k \geq 3$ | 0 | 0 | 0 |

this minimizer, the sharpness is about $1.139855 \times 10^5$, implying a theoretical critical step size for GD of about $1.7546 \times 10^{-5}$.

This experiment follows a design similar to Experiment 1; the detailed setup is provided in the main text. Here, we primarily present additional experimental results. First, Table 3 reports the relative $L^2$ error of the averaged solution across 50 runs for both algorithms. We observe that SGD achieves significantly higher accuracy than GD. Then, in Figure 13, we plot the averaged solution obtained

Table 3: Relative $L^2$ error of the mean function under different algorithms .

| Optimizer | SGD | GD |
|---|---|---|
| Relative $L^2$ Error | 1.231e-06 | 4.081e-03 |

by the two algorithms and overlay the variance band computed from the 50 runs. Finally, we present the variance curves of the solution functions from the 50 runs relative to the averaged solution for both algorithms; see Figure 14 These results clearly indicate that near the global minimizer with high sharpness, SGD can in fact be more stable and yield better performance.

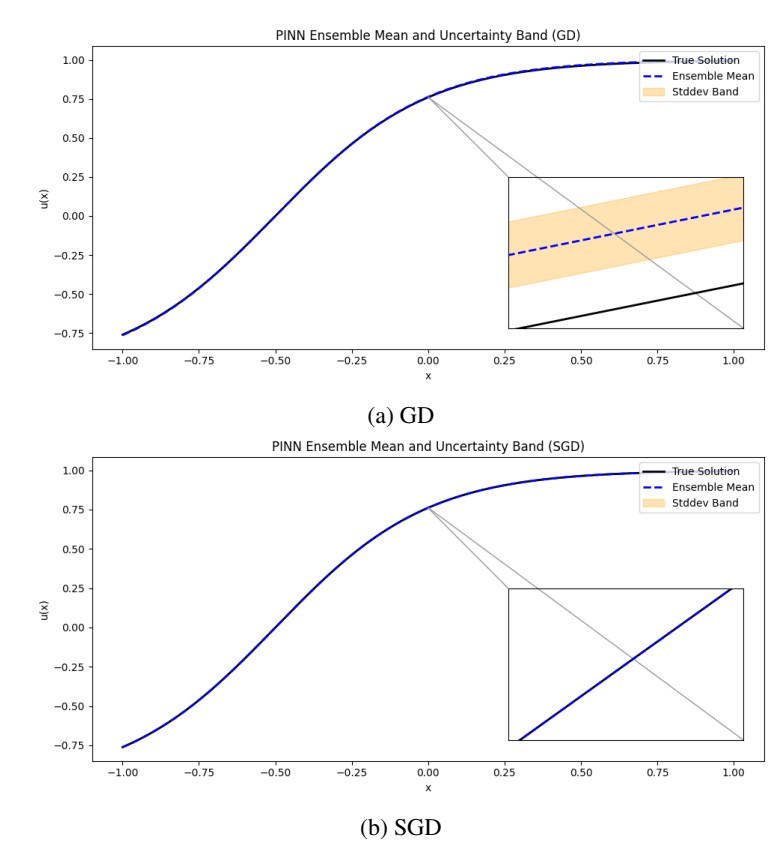

(a) GD

(b) SGD

Figure 13: Averaged solutions and variance bands for SGD and GD over 50 runs.

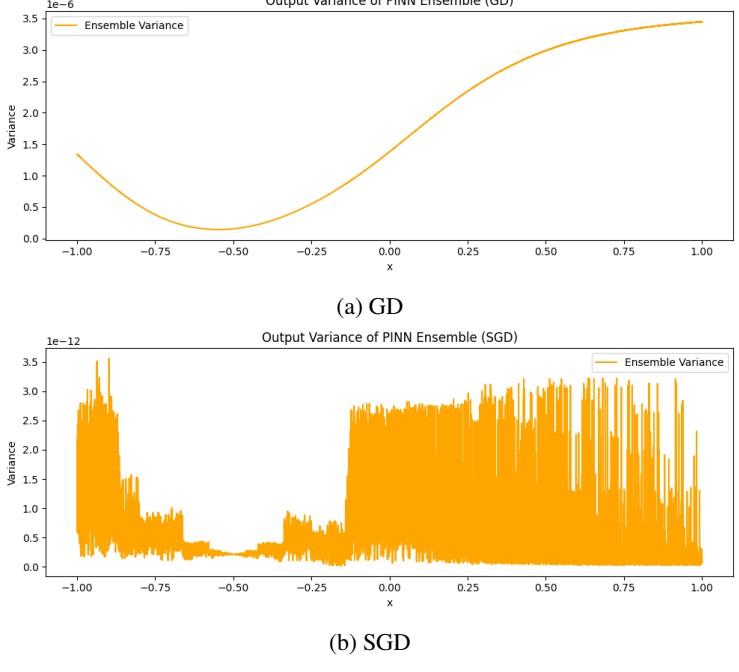

(a) GD

(b) SGD

Figure 14: Variance functions for SGD and GD over 50 runs.

### K.3 EXPERIMENTS ON HELMHOLTZ AND ALLEN–CAHN EQUATIONS

In this subsection, we provide additional numerical results on two prototypical second-order PDEs: the Helmholtz and Allen–Cahn equations. Both equations are considered on the domain $(x, y) \in [-1, 1] \times [-1, 1]$. Their explicit forms are given by

(i) **Helmholtz equation:**

$$\Delta u(x, y) + u(x, y) = f(x, y), \tag{28}$$

(ii) **Allen–Cahn equation:**

$$\Delta u(x, y) - u(x, y)^3 + u(x, y) = f(x, y), \tag{29}$$

where $\Delta$ is the Laplacian operator. In our experiments, we choose the ground-truth solution $u(x, y) = \tanh(x + y + 1)$ and compute the corresponding right-hand side $f(x, y)$ analytically.

We employ a two-layer neural network with width 10, specifically,

$$u_\theta(x, y) = \sum_{j=1}^{10} a_j \tanh(w_j^1 x + w_j^2 y + b_j), \tag{30}$$

where $\theta$ collects all trainable parameters.

For each equation, we utilize the PINN framework to investigate the dynamics of GD and SGD optimizers via three key experiments:

1. We demonstrate that, near a sharpness-moderate minimizer, the step size stability region for SGD is smaller than for GD;

2. We show that, at larger step sizes, SGD instability (i.e., loss explosion) is closely connected to the covariance matrix of the stochastic gradients;

3. We reveal that, in the neighborhood of a highly sharp local minimizer, the step size stability region for SGD becomes larger than for GD.

#### K.3.1 EXPERIMENT 1: STABILITY AT A MODERATE SHARPNESS MINIMIZER

In this experiment, we select a global minimizer $\theta^*$ with parameters $a_1 = 1$, $w_1 = (1, 1)$, $b_1 = 1$ and all other parameters set to zero. Following the logic of Algorithm 1, we perform experiments at various step sizes, with 3000 iterations at each step size. For each PDE, we randomly initialize parameters in the neighborhood of $\theta^*$ and compare the behavior of GD and SGD optimizers.

We observe that, as the learning rate increases, the loss at 3000-th step for SGD exhibits an abrupt jump at a certain threshold, indicating instability of SGD at large step sizes. In contrast, GD remains stable across all tested step sizes. This demonstrates that, near a moderate-sharpness minimizer, the stability region of SGD with respect to the step size is strictly smaller than that of GD.

Figure 15 and Figure 16 present the results for the Helmholtz and Allen–Cahn equations, respectively, each with two independent runs initialized randomly around $\theta^*$.

#### K.3.2 EXPERIMENT 2: COVARIANCE AND INSTABILITY AT LARGE STEP SIZE

In this experiment, we again consider the global minimizer $\theta^*$ with $a_1 = 1$, $w_1 = (1, 1)$, $b_1 = 1$, and all other parameters set to zero. We first analytically compute the critical step size for GD at $\theta^*$. Both GD and SGD are then trained with a learning rate set to $1.5\times$ the theoretical critical step size. During SGD training, we also record the Frobenius norm of the covariance matrix of stochastic gradients at each step.

As shown in Figure 17 and Figure 18, for both the Helmholtz and Allen–Cahn equations, the trajectory of the covariance matrix norm closely aligns with the loss trend in SGD, exhibiting simultaneous increases and instability. This strong correlation suggests that the variance of the stochastic gradients plays a central role in SGD divergence at larger step sizes.

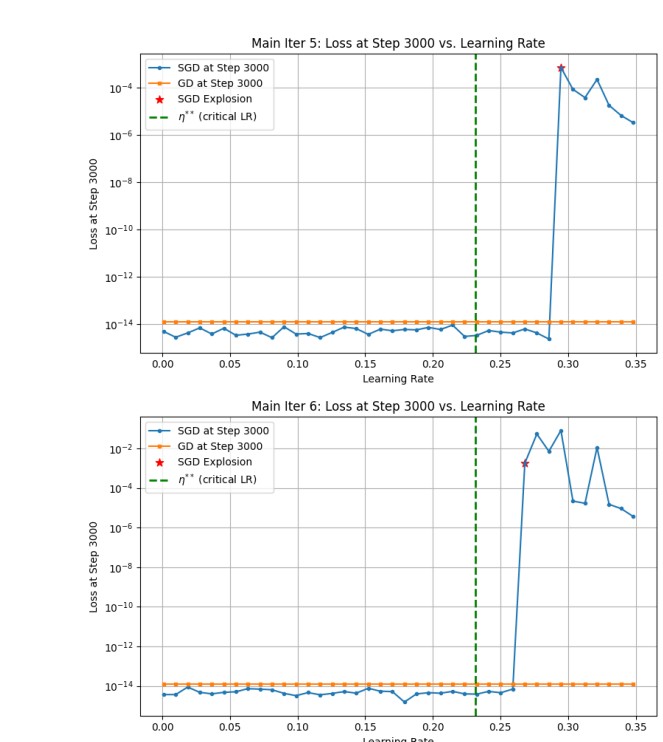

Figure 15: Experiment 1 on the Helmholtz equation: Loss at 3000-th step for various learning rates, under two random initializations near $\theta^*$. Left: Run 1, Right: Run 2.

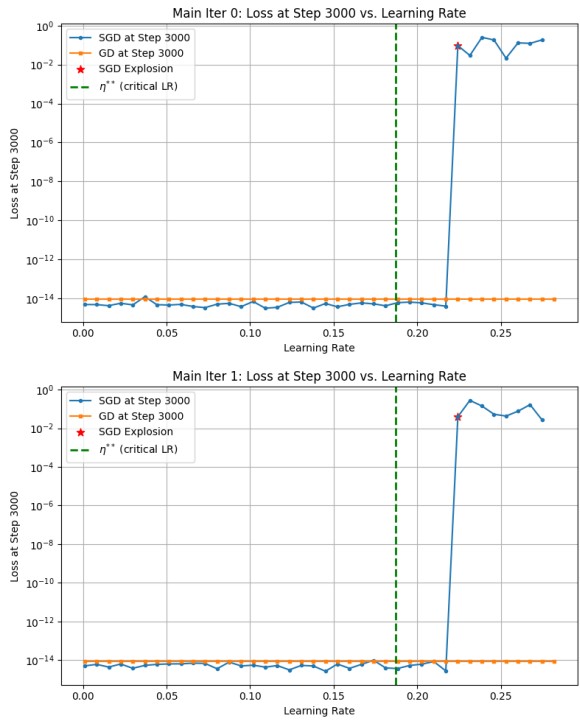

Figure 16: Experiment 1 on the Allen–Cahn equation: Loss at 3000-th step for various learning rates, under two random initializations near $\theta^*$. Left: Run 1, Right: Run 2.

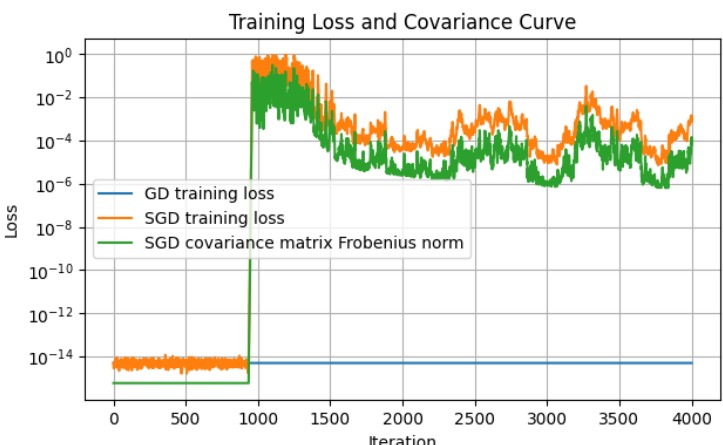

Figure 17: Experiment 2 on the Helmholtz equation: The loss and covariance matrix show highly consistent rising trends and instability.

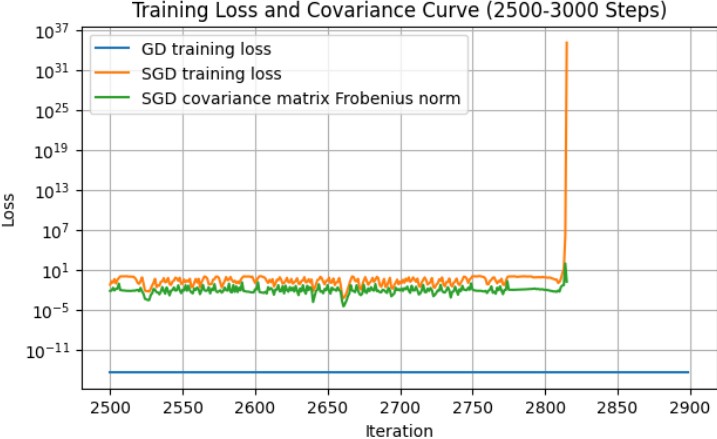

Figure 18: Experiment 2 on the Allen–Cahn equation: The evolution of the covariance norm mirrors that of the loss, supporting the key role of gradient variance in SGD instability.

### K.3.3 EXPERIMENT 3: STABILITY AT A HIGHLY SHARP MINIMIZER

In this experiment, we focus on a minimizer $\theta^{**}$ of higher sharpness, where the parameter values are set as follows: $a_1 = 51$, $w_1 = (1,1)$, $b_1 = 1$, $a_2 = -25$, $w_2 = (1,1)$, $b_2 = 1$, $a_3 = -25$, $w_3 = (1,1)$, $b_3 = 1$, and all remaining parameters are zero. Around this point, we perform 20 independent random initializations. For each run, we first calculate the theoretical critical step size for GD at $\theta^{**}$, then train PINNs with GD and SGD using a step size exceeding the critical value by $10^{-6}$ for 3000 iterations.

We observe that, across all runs for both the Helmholtz and Allen–Cahn equations, the GD loss typically exhibits a sudden increase at the beginning, followed by a very slow decrease, while SGD achieves rapid loss reduction and converges much faster. By the end of training, the final losses of SGD are often several orders of magnitude lower than those of GD. This indicates that, in the vicinity of highly sharp minimizers, the stochasticity in SGD can significantly enhance convergence properties compared to GD, even when the step size slightly exceeds the classical stability threshold for GD. Figure 19 and Figure 20 present representative loss curves from two random initializations for each equation.

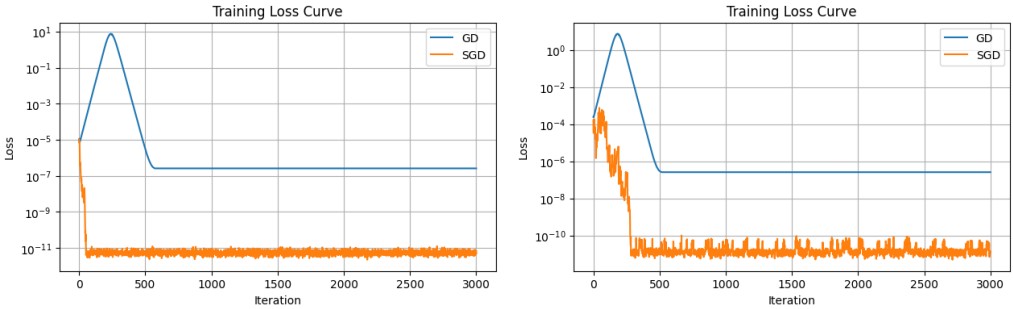

Figure 19: Experiment 3 on the Helmholtz equation: Loss curves of GD and SGD using a step size just above the critical threshold, for two random initializations near $\theta^{**}$.

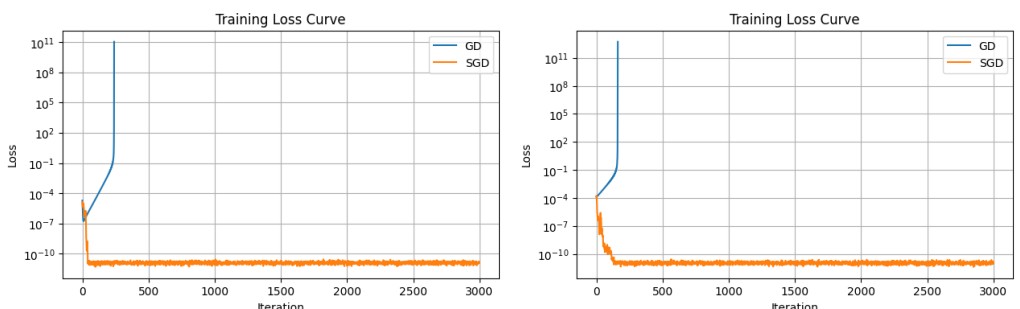

Figure 20: Experiment 3 on the Allen–Cahn equation: Two illustrative runs.

