# OpenReview forum: "Modeling Training Dynamics and Error Estimates of DNN-based PDE Solvers: A Continuous Framework"
_ICLR.cc/2026/Conference — Submitted to ICLR 2026_

### Official Review · Reviewer_5Xb2 · 2025-10-29

**Soundness:** 1
**Presentation:** 3
**Contribution:** 2
**Rating:** 4
**Confidence:** 4

**Summary:**

The paper claims to propose a continuous-time theoretical framework for neural PDE solvers that removes the need for the global Lipschitz continuity assumption. It introduces a noisy high-order regularization to ensure well-posedness of an SDE corresponding to discrete SGD, proving local weak convergence and providing long-term error estimates for DNN-based PDE solvers. The theoretical contribution is novel, but relies on strong assumptions. Its practical impact is also quite unclear.

**Strengths:**

- Theoretical novelty: proposes a continuous-time analysis framework for neural PDE solvers without global Lipschitz assumptions.
- Provides long-term error estimates and Laplace-type approximations that may be useful for understanding training dynamics.
- Establishes local weak convergence from discrete SGD to a regularized SDE.

**Weaknesses:**

While theoretically interesting, the strong assumptions and limited experimental validation reduce the practical impact of the work. The main concerns is listed as following
- Very limited baseline comparisons. Modern optimizers like AdamW or other recent related literatures (for example **Newton Informed Neural Operator for Solving Nonlinear Partial Differential Equations**) are not compared in the experiments. Based on the results in the manuscript, it's difficult to claim better convergence performance.
- The theoretical derivation has too many restrictions which can significantly reduce the applicability of the proposed method.

**Questions:**

- The framework relies on a $|θ|^{2s}$ (s≥10) term to guarantee well-posedness. Such assumption may not be desirable in many application conditions.
- Convergence is guaranteed only in bounded regions and depends on the probability of the process staying within that region. Its global convergence in high-dimensional or steep PDE loss landscapes is not addressed properly.
- Too simple test cases: Only 2D ODEs / low-dimensional PDEs are tested with very small networks (2 layers, width 10). Performance on complex, nonlinear, high-dimensional PDEs or real physical systems is unknown.
- Experiments compare only SGD and GD, without modern optimizers (Adam, AdamW, L-BFGS, etc.), making the practical advantage of the proposed approach unclear.
- Only L2 errors and loss curves are reported. While convergence speed, training time, generalization, and robustness are not evaluated thoroughly.

---

> ### Author Response · Authors · 2025-11-20
>
> Thank you for your thorough review and for highlighting the main contributions and strengths of our work. We appreciate your recognition of our theoretical framework and the value of our error estimates and convergence results. Below, we address your questions and concerns in detail.
>
> ## Replies to Weaknesses
>
> >### Weakness 1:
> >
> > Very limited baseline comparisons...
>
> ### Reply to Weakness 1:
>
> Thank you for this valuable comment. As you summarized in your review, the main contribution of our paper is to remove the global Lipschitz assumption and to provide a well-posed continuous-time SDE approximation for the SGD dynamics of neural PDE solvers. Our proposed algorithm—the noisy regularized SGD—is designed to ensure that the corrected SDE remains well-posed, even when the drift is not globally Lipschitz. It is not intended to introduce a practically superior or faster-converging new optimizer. Therefore, our experiments (see Section I) are mainly to illustrate that the regularized algorithm can still solve certain problems in practice and, for small regularization and noise, behaves similarly to standard SGD. There is no necessity to demonstrate superior convergence speed over AdamW or the optimizer in your cited works.
>
> Additionally, we have added experiments as in Section 4 in the PDE setting to further support our claims in that section (please see Appendix K).
>
> Finally, if you are interested in continuous-time approximations for algorithms like AdamW or other optimizers, our framework can be adapted: one can modify the algorithm, check if the associated SDE admits a Lyapunov function to guarantee well-posedness, and then study the weak error between the discrete and continuous-time dynamics.Thank you again for this suggestion.
>
> > ### Weakness 2:
> >
> > The theoretical derivation has too many restrictions....
>
> ### Reply to Weakness 2:
>
> Thank you for your question. Indeed, our theoretical analysis relies on Assumptions 1–3. The first two are standard in the literature, so we would like to further clarify the meaning and necessity of Assumption 3.
>
> On one hand, Assumption 3 is actually weaker than requiring a global Lipschitz loss gradient. If the loss gradient is globally Lipschitz, one can show—using arguments similar to those in [1]—that Assumption 3 automatically holds.
>
> On the other hand, this is currently the best we can do for global results. Without Assumption 3, Proposition 1 still provides first-order weak convergence of the noisy regularized SGD to its SDE on bounded regions. Assumption 3 is mainly used to estimate the probability that trajectories escape from these regions, enabling us to present global weak error estimates (Theorem 1). In practical training, methods such as gradient clipping can help prevent trajectories from blowing up, so weak convergence essentially still holds. This assumption is also commonly adopted in the related literature.
>
> Additionally, we have added experimental validation of this assumption in the paper (see Appendix E for details).
>
> Thank you again for highlighting this point. In future work, we will consider how to further relax Assumption 3.
>
> ## Replies to Questions
>
> >### Question 1:
> >
> > The framework relies on a $\|\theta\|^{2s}$ ($s \geq 10$) regularization term....
>
> ### Reply to Question 1:
>
> Thank you for your question. We agree that this type of high-order regularization is not commonly used in practical training. However, this is the minimal regularization needed to rigorously guarantee the well-posedness of our SDE. Without this term, the drift of the SDE is neither globally Lipschitz nor does it admit a suitable Lyapunov function (see Lemma 3), which means that the SDE may not have a solution in theory. We prefer not to present theoretically incomplete or unrigorous results, so we carefully introduce this regularization and construct a Lyapunov function to ensure the well-posedness of the corrected SDE, which is essential for our subsequent weak error analysis.
>
> Intuitively, although this regularization has a high order, when the regularization coefficient tends to zero, the trajectory of the regularized SGD closely approximates that of the original SGD. Thus, the corrected SDE can still serve as a good theoretical approximation. Our introduction of this regularization is not meant to suggest its practical use, but rather as a theoretical tool.
>
> Furthermore, most previous works obtain SDE approximations by directly assuming globally Lipschitz loss gradients, which is only satisfied by very simple network architectures (such as linear, e.g., random feature models). In contrast, our approach requires adding a regularization term to the loss, but the resulting theory applies to practical network architectures, which rarely satisfy the global Lipschitz condition even in simple settings like two-layer fully connected networks.
>
> Thank you again for raising this important point.

---

> ### Author Response · Authors · 2025-11-20
>
> >### Question 2:
> >
> > Convergence is guaranteed only in bounded regions and depends on the probability of the process staying within that region. Its global convergence in high-dimensional or steep PDE loss landscapes is not addressed properly.
>
> ### Reply to Question 2:
>
> Thank you for your question, and your understanding is correct.
>
> Indeed, in overly steep loss landscapes, Assumption 3 may not hold. In this case, we can only establish Proposition 1, which provides first-order weak convergence within bounded regions, and a global weak error estimate is not available. However, in practice, various techniques such as gradient clipping are often used to enhance training stability. Such techniques can help keep the trajectory within a bounded region, so that the result in Proposition 1 essentially applies.
>
> Moreover, without a globally Lipschitz loss gradient, global weak convergence is generally unattainable. We have tried to strike a balance between theoretical rigor and practical considerations in our analysis. Thank you again for pointing out this issue.
>
> >### Question 3:
> >
> >Too simple test cases: Only 2D ODEs / low-dimensional PDEs are tested with very small networks (2 layers, width 10). Performance on complex, nonlinear, high-dimensional PDEs or real physical systems is unknown.
>
>
> ### Reply to Question 3:
> Thank you for your comment. In response, we have added an experiment on the Helmholtz and Allen--Cahn equations, and found results that are consistent with those reported in Section 4. Specifically, we observe that near minimizers with low sharpness, GD demonstrates greater stability with respect to the step size compared to SGD, while the opposite is true near minimizers with very high sharpness. The detailed experimental setup and results are now provided in the appendix.
>
> >### Question 4:
> >
> > Experiments compare only SGD and GD, without modern optimizers (Adam, AdamW, L-BFGS, etc.), making the practical advantage of the proposed approach unclear.
>
> ### Reply to Question 4:
>
> Thank you for your comment. As we explained in our response to Weakness 1, our goal is not to propose a new optimizer or to promote the practical advantage of the noisy regularized SGD algorithm. Our main contribution is to provide a framework that establishes a well-posed continuous-time SDE approximation for discrete stochastic optimization algorithms when the loss gradient is not globally Lipschitz.
>
> The experiments in Section 4 are designed to compare SGD and GD only to illustrate the effect of stochasticity and the applicability of our theoretical framework. A comparison with Adam or other optimizers is not necessary for the main objectives of this work.
>
> Thank you again for your thoughtful feedback.
>
> >### Question 5:
> >
> > Only L2 errors and loss curves are reported. While convergence speed, training time, generalization, and robustness are not evaluated thoroughly.
>
> ### Reply to Question 5:
>
> Thank you for your question.
>
> The experiments in our paper serve two main purposes. The first is to demonstrate that, when the regularization coefficient and noise level are small, the performance of the noisy regularized SGD is similar to that of standard SGD. The second is to illustrate the effects of the stochasticity introduced by mini-batch noise in SGD.
>
> The metrics we reported—L2 errors and loss curves—are chosen to directly support these aims. Other aspects such as training time, generalization, and robustness are not primary concerns of this work, as our focus is on providing a theoretical framework rather than proposing or recommending a new practical algorithm. The algorithms we use are basic and well-understood, so additional metrics like training time are less relevant in our setting.
>
> As noted previously, the noisy regularized SGD is introduced as a theoretical tool rather than an optimizer to be adopted in practical settings, and our experiments are designed accordingly.
>
> Additionally, our code is provided as supplementary material, so interested readers can check further experimental details if desired.
>
> Thank you very much for your careful reading and thoughtful feedback on our work. We have addressed each of your questions and comments in detail, and we sincerely appreciate your suggestions, which have helped us clarify the objectives and contributions of our paper. If you have any further questions or would like to discuss related topics, we would be glad to continue the discussion. Thank you again for your valuable suggestions.
>
> [1] GNMil’shtein Weak approximation of solutions of systems of stochastic differential equations. Theory of Probability & Its Applications, 30(4):750–766, 1986.

---

> ### Comment · Reviewer_5Xb2 · 2025-11-25
>
> Thank you for the additional experiments and for strengthening the manuscript. These updates make the paper clearer and more complete. However, my main concern regarding the motivation remains only partially addressed.
>
> Relaxing the global Lipschitz assumption is mathematically meaningful and could, in principle, benefit methods such as PINNs or other neural PDE solvers. Nevertheless, in its current form, the proposed “noisy regularized SGD” is presented primarily as a theoretical construct, and the experiments—while illustrative—do not demonstrate practical advantages in training efficiency, accuracy, or robustness over standard optimizers.
>
> Overall, while the theoretical contribution is appreciated, its practical impact remains unclear, and the motivation does not yet fully align with the applied AI focus of this venue. I will maintain my current evaluation unless further contributions are made on the AI side in this manuscript.

---

### Official Review · Reviewer_8rio · 2025-10-30

**Soundness:** 3
**Presentation:** 3
**Contribution:** 2
**Rating:** 4
**Confidence:** 3

**Summary:**

This paper develops a continuous-time framework based on SDEs to analyze the training dynamics of deep neural network-based PDE solvers, specifically PINNs. The authors introduce a "noisy regularized SGD" algorithm that adds both a high-order regularization term and Gaussian noise to standard SGD. They establish weak convergence between this discrete algorithm and its continuous SDE approximation, removing the restrictive global Lipschitz assumption common in prior work. The paper also characterizes asymptotic error via the SDE's invariant measure using WKB approximation and the Laplace method. Experiments on a simple 1D ODE reveal that stochasticity narrows the stability regime of learning rates and degrades solution accuracy compared to gradient descent, though SGD can outperform GD near sharp minima.

**Strengths:**

1. The most significant theoretical contribution of this paper is to establish weak convergence results without requiring global Lipschitz continuity of the loss function. As the authors note, "our theory more applicable to practical deep networks" since standard neural networks violate this assumption. The local Lipschitz approach using stopping times (Definition 2) and the decomposition in Theorem 1 is technically sound.
2. The perspective of analyzing long-term error through the invariant measure of the SDE (Proposition 4) and its asymptotic approximation via WKB methods is interesting. The connection between the maximizers of $S_0$ and the expected error in Proposition 5 provides a different lens on understanding SGD behavior beyond standard optimization/generalization decompositions.
3. The experiments in Section 4 provide valuable empirical insights. The observation that SGD has a much narrower stability domain than GD near low-sharpness minima (Figure 1) and that stochasticity degrades accuracy even within the stable regime helps explain why PINNs often struggle with precision. The comparison between two regimes with different sharpness levels is well-motivated.
4. The paper is generally well-written with careful definitions and the proofs appear rigorous.

**Weaknesses:**

1. The practical applicability of the modified algorithm is a bit limited. The noisy regularized SGD requires an extremely high-order regularization term to ensure theoretical guarantees. While the authors acknowledge this in Remark 4, the gap between theory and practice can be concerning. The experiments in Appendix I show the algorithm works on some problems, but the regularization parameter and noise level appear highly problem-dependent. It's unclear how practitioners should set these hyperparameters or whether the theoretical insights transfer to standard SGD.
2. The uniform moment bounds in Assumption 3 are crucial for Proposition 3 and Theorem 1, yet the authors simply assume this holds without proof or empirical verification. They acknowledge it "remains open in many settings," which significantly weakens the main result. Without this, the weak convergence is only established for processes that stay in $B_R$, and the exit probability bounds don't apply uniformly in $\eta$.
3. The experimental validation focuses exclusively on a simple 1D ODE with width-10 networks where the exact solution can be represented. While this enables precise analysis, it's far from the high-dimensional PDEs that motivate DNN-based solvers. The experiments in Appendix I on 2D problems (Helmholtz, Fisher-KPP, Allen-Cahn) show the algorithm can work but don't validate the theoretical predictions about stochasticity effects or compare against standard methods systematically.
4. The paper promises "actionable guidance for practical training" but the main takeaway is not too surprising and doesn't lead to clear recommendations. The suggestion that "adaptively switching optimizers and step sizes...can be beneficial" is vague and not demonstrated.

**Questions:**

1. Can the authors provide any theoretical or empirical evidence to support Assuption 3? At minimum, is it possible to verify whether it holds for the experimental settings in Section 4?
2. The WKB approximation assumes $B_0 \in C^2$, but is this regularity guaranteed, given that the loss landscape of neural networks is typically not smooth?
3. In regime 2, SGD outperforms GD even when using a learning rate that causes GD to diverge. Does this contradict the stability analysis?
4. This paper https://proceedings.mlr.press/v235/chen24ad.html proposes a method that the author claims to solve Allen-Cahn equation (and other equations) towards machine precision. Can the authors comment on how the framework proposed in this paper is consistent with the method in https://proceedings.mlr.press/v235/chen24ad.html ? Is the framework not applicable in their setting at all? If not, can a similar framework be developed?
5. Are the theoretical results sensitive to the choice of $s=10$ in the regularization? The proof is for $s\ge10$, but is there an intuition for why?

---

> ### Author Response · Authors · 2025-11-20
>
> Thank you for your careful review and thoughtful summary of our work. We appreciate your recognition of both the theoretical and empirical contributions, as well as your positive feedback on our analysis and presentation. Below, we respond to your specific questions and concerns point by point.
>
> ## Replies to Weaknesses
>
> >### Weakness 1:
> >
> > The practical applicability of the modified algorithm is a bit limited....
>
> ### Reply to Weakness 1:
>
> Thank you for your question.
>
> The core motivation of our work is to provide a rigorous continuous-time approximation for SGD in the context of PDE solvers, where the loss function is often highly complex and does not have globally Lipschitz gradients. In such settings, the associated SDE typically lacks globally Lipschitz drift, making it ill-posed and preventing the application of existing theoretical results.
>
> To address this, we introduce the noisy regularized SGD as a *theoretical auxiliary algorithm*—the added regularization and noise terms grant the SDE good properties, such as uniformly positive definite diffusion, which ensure well-posedness even without global Lipschitz assumptions. This approach allows us to break through the major limitation of past work that always required globally Lipschitz gradients.
>
> Importantly, the noisy regularized SGD is not intended as a new practical algorithm, but rather as a tool for theory. In practice, as the regularization and noise parameters approach zero, the iterates of this modified algorithm converge to those of the standard SGD. We also quantify the weak error between the modified algorithm and the corresponding SDE, justifying the interpretation of the regularized SDE as a continuous-time approximation of SGD.
>
> We acknowledge that tuning these hyperparameters in practice is difficult and the modified algorithm itself may not be practical. Its main purpose is to ensure theoretical rigor. Thank you again for the opportunity to clarify this point.
>
> >### Weakness 2:
> >
> > The uniform moment bounds in Assumption 3 are crucial for Proposition 3 and Theorem 1, yet the authors simply assume this holds without proof or empirical verification....
>
> ### Reply to Weakness 2:
>
> Thank you for this important point. We have added both empirical evidence and additional explanations of this assumption in the revised manuscript. Below, we provide clarification regarding the reasonableness and necessity of Assumption 3:
>
> **1. Rationale:**
> Assumption 3 is essentially milder than the global Lipschitz loss gradient. If the loss has a globally Lipschitz gradient, this assumption can be established following the proof technique in Lemma 5 of [1]. Therefore, it is a classical and reasonable technical assumption, even though it is not trivial to establish in general.
>
> **2. Necessity and Role:**
> Currently, this is the weakest assumption under which we can provide global weak error estimates. Without it, we can still establish Proposition 1—i.e., weak convergence between noisy regularized SGD and the associated SDE—but only within bounded regions. The uniform moment bound is crucial for controlling the probability of the process escaping these regions, which is necessary for the global error estimate in Theorem 1. In practice, stabilization techniques like gradient clipping further support this assumption by helping keep the process within a bounded domain. Moreover, this is a standard assumption in recent related works [2].
>
> **3. Empirical Verification:**
> We have added experimental results to empirically verify this assumption. Specifically, we performed training runs with 40 different step sizes ranging from $10^{-5}$ to $10^{-3}$, each for $1/\eta$ steps. For each case, we recorded the maximal squared $\ell^2$-norm of the parameters during training. These maxima are plotted against the step size in the appendix E. The results show that larger maxima are observed as the step size approaches $10^{-5}$, suggesting that verifying this assumption focuses primarily on sufficiently small step sizes.
>
> On a theoretical note, for fixed $p$ and $\eta$, the SGD iterates satisfy:
> $$
> \mathbb{E}|\theta_{k+1}|^{2p} \leq \mathbb{E}|\theta_k|^{2p} + C\eta \left(1 + \mathbb{E}|\theta_k|^{2p + 2(s-1)}\right).
> $$
> This indicates that as $\eta \to 0$, the discrete dynamics are effectively controlled by the following ODE:
> $$
> \frac{du}{dt} = C\left(1 + u^{1+\frac{1}{p}(s-1)}\right),
> $$
> for which classical theory guarantees uniform boundedness on $[0, T]$ for some $T > 0$. Thus, uniform moment bounds hold for small enough step sizes.
>
> This also reflects a broader point: mathematical tools for discrete-time systems are generally less developed compared to continuous-time theory, motivating our work to facilitate understanding of SGD using SDE approximations.
>
> Once again, thank you for your insightful question. We plan to further investigate conditions for weak error estimates that require even weaker assumptions than Assumption 3 in future work.

---

> ### Author Response · Authors · 2025-11-20
>
> >### Weakness 3:
> >
> >The experimental validation focuses exclusively on a simple 1D ODE with width-10 networks where the exact solution can be represented.
>
> ### Reply to Weakness 3:
>
> Thank you for your comment. In response, we have added an experiment involving the Helmholtz and Allen--Cahn equations, and the results are consistent with those presented in Section 4. Specifically, we find that near minimizers with low sharpness, GD demonstrates greater robustness to step size choices than SGD, while the opposite trend is observed near minimizers with very high sharpness. The details of the experimental setup and full results are now included in the appendix.
>
> Our initial focus on problems with closed-form solutions was primarily for convenience, as this allows us to explicitly select and systematically analyze specific minimizers. Prior work [3] has shown that the step size stability region for SGD is typically smaller than that for GD, which agrees with our findings near less sharp minimizers. We expect this property to extend to more general PDE settings as well.
>
> Interestingly, our observation that SGD appears more stable than GD around highly sharp minimizers diverges from established conclusions. We find this phenomenon noteworthy and plan to investigate it further in future work.
>
> Thank you again for bringing up this important point.
>
> >### Weakness 4:
> >
> >The paper promises "actionable guidance for practical training" but the main takeaway is not too surprising and doesn't lead to clear recommendations. The suggestion that "adaptively switching optimizers and step sizes...can be beneficial" is vague and not demonstrated.
>
> ### Reply to Weakness 4:
>
> Thank you for your thoughtful comments. The primary contribution of our paper is to provide a rigorous theoretical foundation and continuous-time modeling framework for understanding the training dynamics of neural network-based PDE solvers. We believe that a continuous-time perspective allows access to a richer set of theoretical tools and insights than what is currently available for discrete-time systems. In prior work [4, 5], SDE-based models combined with optimal control theory have even led to specific, actionable strategies such as optimal batch size scheduling, which has influenced algorithmic designs. Extending in this direction remains an interesting avenue for future research.
>
> In addition, we present results in Appendix F of the original manuscript that highlight various properties of the SDE solutions, offering deeper insights into the typical behavior of SGD training trajectories in practice. While these findings are primarily theoretical, they may serve as useful guiding principles for practitioners.
>
> Regarding your point about the phrase "adaptively switching optimizers and step sizes...can be beneficial," we agree that our current discussion of this idea is rather general, reflecting only an observation from the experimental phenomena in Section 4, without dedicated experiments or systematic validation. Nonetheless, we note that switching optimizers at different training stages is already a common practical strategy when training PINNs and related models. We appreciate your suggestion and will clarify this point, while treating it as a possible hypothesis for future work rather than a demonstrated result.
>
> Once again, thank you for your valuable feedback, which will help us refine the claims and presentation of the paper.
>
> ## Replies to Questions
>
> >### Question 1:
> >
> >Can the authors provide any theoretical or empirical evidence to support Assuption 3?
>
> ### Reply to Question 1:
> Please see our response to Weakness 2.
>
> >### Question 2:
> >
> > The WKB approximation assumes $S_0 \in C^2$, but is this regularity guaranteed, given that the loss landscape of neural networks is typically not smooth?
>
> ### Reply to Question 2:
>
> Thank you for your question. In our setting, we use the $\tanh$ activation function, which generally ensures that the resulting loss function is smooth. More importantly, the $S_0$ in our analysis is the leading-order term in the expansion of the stationary distribution $p$ to the SDE. As shown in Proposition 4, $p$ satisfies a second-order PDE, which implies $p \in C^2$ and thus the assumption $S_0 \in C^2$ is reasonable under standard regularity theory. Therefore, the assumption $S_0 \in C^2$ relies on the regularity of $p$, not directly on the global smoothness of the loss landscape. Thank you again for your thoughtful comment.

---

> ### Author Response · Authors · 2025-11-20
>
> >### Question 3:
> >
> > In regime 2, SGD outperforms GD even when using a learning rate that causes GD to diverge. Does this contradict the stability analysis?
>
> ### Reply to Question 3:
>
> Thank you for this question. As we mentioned in the paper, this is indeed a somewhat surprising phenomenon, and we agree that it appears not to fully align with standard stability analysis. Intuitively, at local minimum with very high sharpness, certain directions in the loss landscape may cause gradient descent to diverge, while the stochasticity in SGD might allow the optimizer to occasionally find more favorable directions or escape from problematic areas. We believe this effect deserves further investigation and will be a focus of our future theoretical analysis. Thank you again for pointing this out.
>
> >### Question 4:
> >
> > This paper proposes a method that the author claims to solve Allen-Cahn equation (and other equations) towards machine precision....
>
> ### Reply to Question 4:
> Thank you for bringing up this interesting reference. We have carefully read the paper (Chen et al., 2024) and acknowledge that it presents an excellent method—TENG—that achieves remarkable accuracy for solving Allen-Cahn and other evolution equations. The methodological contribution of their work is significant, and we appreciate the opportunity to clarify the relationship between their approach and ours.
>
> However, our framework is fundamentally different in its focus and main contribution. Our work is concerned with the theoretical continuous-time modeling and analysis of the *training dynamics* of deep learning-based solvers for PDEs, specifically through SDE approximations of the optimization (training) process itself. In contrast, Chen et al.'s paper proposes a novel algorithm for *solving* evolution PDEs, aiming for high-precision solutions.
>
> If one seeks a direct connection, a possible intersection could lie in the internal training process for each parameter $\theta_t$ at every time step within the TENG method. The dynamics of such training could, in principle, be analyzed using our continuous-time framework to gain insight into the optimization behavior or to guide algorithmic design. Nevertheless, the main objectives and scopes of the two works remain substantially distinct.
>
> Thank you again for giving us the chance to clarify the relationship between the two approaches.
>
> >### Question 5:
> >
> > Are the theoretical results sensitive to the choice of $s=10$ in the regularization? The proof is for $s \geq 10$, but is there an intuition for why?
>
> ### Reply to Question 5:
>
> Thank you for your question.The main purpose of introducing the high-order regularization term is to construct a valid Lyapunov function for the SDE with a non-globally Lipschitz drift (see Lemma 3 in the Appendix), which ensures that the noisy regularized SDE is well-posed. As long as $s \geq 10$, Lemma 3 holds, and therefore all the theorems in our paper follow. The specific value of $s$ above this threshold does not affect the validity of the results.
>
> Intuitively, previous works always assume a globally Lipschitz loss gradient, making the SDE drift both Lipschitz and of linear growth, which guarantees existence and uniqueness of the solution by classic SDE theory. In our setting, without the global Lipschitz gradient, the drift can grow polynomially. By adding a sufficiently high-order regularization term (at least $|\theta|^{2s}$ for $s\ge 10$), we can control the growth of the drift, establish a Lyapunov function, and make the SDE well-posed. Thank you again for this insightful question.
>
> Thank you very much for your valuable comments and for taking the time to review our manuscript. We appreciate your insightful suggestions, which have allowed us to further explain and improve our work. We hope our responses have addressed your concerns. Please feel free to reach out with any additional questions or suggestions—we welcome ongoing discussion and feedback. Thank you again for your support and careful consideration.
>
> [1] GN Mil’shtein. Weak approximation of solutions of systems of stochastic differential equations. Theory of Probability & Its Applications, 30(4):750–766, 1986.
>
> [2] Zhongwang Zhang, Yuqing Li, Tao Luo, and Zhi-Qin John Xu. Stochastic modified equations and dynamics of dropout algorithm. In The Twelfth International Conference on Learning Representations, 2024.
>
> [3] Lei Wu, Chao Ma, and Weinan E. How SGD selects the global minima in over-parameterized learning: A dynamical stability perspective.Advances in Neural Information Processing Systems, 31, 2018.
>
> [4] Qianxiao Li,Cheng Tai, and Weinan E. Stochastic modified equations and adaptive stochastic gradient algorithms. In International Conference on Machine Learning, pp.2101–2110. PMLR, 2017.
>
> [5] Jing An, Jianfeng Lu, and Lexing Ying. Stochastic modified equations for the asynchronous stochastic gradient descent. Information and Inference: A Journal of the IMA 9.4 (2020): 851-873.

---

### Official Review · Reviewer_Bqfp · 2025-10-31

**Soundness:** 1
**Presentation:** 3
**Contribution:** 2
**Rating:** 2
**Confidence:** 3

**Summary:**

This work contributes to a deeper understanding of the optimization procedure with stochastic gradient descent (SGD) using tools from stochastic differential equations (SDEs) in the case of training feed-forward neural networks for a class of partial differential equations. The estimation error of the iterative SGD algorithm compared to its continuous counterpart is characterized in the weak form. Computational experiments are used to investigate the effect of stochasticity on the stability of the optimization and provide a comparison to classical gradient descent.

**Strengths:**

The paper makes use of established connection of SGD and SDE via stochastic modified equations (SME) (Li et at. 2017, 2019) and investigates the consequences of modified assumptions. So far this community has not focused much on neural networks for partial differential equations, which is a part of the novelty of this work. For the optimizer setting the authors consider, including the new assumptions, they find that the optimizer weakly converges to its continuous SDE version. An important result is the relaxation of Lipschitz continuity assumption on the network, which was used by previous works.
The experimental results are consistent with existing literature on the dynamics of SGD optimizers. To the best of my knowledge, the results are novel for the considered noisy regularized SGD optimizer.

**Weaknesses:**

1) The authors position this contribution in the realm of learning methods for general partial differential equations (PDEs), but it seems that the focus is on specific, second-order elliptic PDEs, as stated in section 2.1.. I suggest making this clear at least in the abstract and the title. The main results (theorem 1) heavily rely on this specific PDE type, with assumptions (1) for the coefficient terms.
2) "... a continuous framework" is also too generic for the title (use "continuous-time" instead?); it is also challenging to read about "continuous time" frameworks while the PDEs in question are not time-dependent, yet the introduction and abstract make no distinction about this.
3) The authors claim that "..our results readily extend to more general equations and deeper network architectures", but do not prove any of this, or demonstrate it experimentally. The same seems true for "all results extend directly to the empirical setting" (l155), where it is not clear at all why choosing a finite-size data set (with e.g. only 10 evaluation points, in higher base-space dimensions... etc) would result in "direct" extension from the exact L^2 error.
4) Similarly, in the title and the main text, "dnn-based PDE solvers" are mentioned, but in the setting (e.g. eq. 2), only shallow, two-layer networks are considered. Merely stating that "the results readily extend to deeper architectures" is not enough to use a broad statement in the title about DNNs. Shallow networks behave very differently from deep networks, and have very different properties (see e.g. "Poole, Ben, Subhaneil Lahiri, Maithra Raghu, Jascha Sohl-Dickstein, and Surya Ganguli. 2016. "Exponential Expressivity in Deep Neural Networks through Transient Chaos." In Advances in Neural Information Processing Systems 29, edited by D. D. Lee, M. Sugiyama, U. V. Luxburg, I. Guyon, and R. Garnett. Curran Associates, Inc. http://papers.nips.cc/paper/6322-exponential-expressivity-in-deep-neural-networks-through-transient-chaos.pdf".)
5) The computational experiments do not show the performance of SGD on a PDE, but just an ODE. There is no discussion why this is a good test case for behavior on PDEs, beyond "it has a closed form solution".
6) For many PINNs the optimizer starts with Adam iterations, followed by additional iterations with L-BFGS (a quasi-Newton method), this is not acknowledged and should be mentioned either as a limitation or future work, especially as the abstract mentions "adaptively switching optimizers".
7) The manuscript lacks a reflection on the limitations of the current work.

8) Minor remarks:
- The text in the figures is too small
- It seems that the same notation $|\cdot|$ is used both for the $L^2$ vector norm as well as for the determinant. Ideally two different notations would be used (e.g. $\|\cdot\|$ for norm, and normal lines $|$ for determinant, or use $\det(\cdot)$).
- Several times "continuous modeling" is used (even in the title, and l.053, for example), while I presume the authors meant "continuous-time modeling". This must be changed.
- The explanation of regularization of powers 20 is not sufficient (l240). It indeed seems extremely artificial, and it is not clear at this point in the paper why the power 20 is the barrier between unbounded and bounded growth. There should at least be a link to a discussion later in the paper, or the proof where this is clarified.

**Questions:**

1. Other work (Li et al. 2017, 2019) considers not only SGD but also SGD with momentum and Nesterov's method. Is an extension of your work possible for these variants of SGD?
2. The loss function for PINNs often includes additional terms (e.g. boundary/initial conditions, additional physical constrains, enforcing symmetry), what kind of impact would these terms have on the SGD dynamics, or what assumptions should be made in regard to them?

---

> ### Author Response · Authors · 2025-11-20
>
> Thank you very much for your thoughtful review and for highlighting the key contributions of our work. We truly appreciate your engagement with our paper. We also noticed that there may be some misunderstandings or points that were not clearly conveyed in our original submission. In the following, we address your concerns and clarify each point in detail.
>
> ## Replies to Weaknesses
>
> > ### Weakness 1:
> >
> > The authors position this contribution in the realm of learning methods for general partial differential equations (PDEs), but it seems that the focus is on specific, second-order elliptic PDEs, as stated in section 2.1....
>
> ### Reply to Weakness 1:
>
> Thank you for raising this important point. We apologize for any confusion our presentation may have caused regarding the generality of our results.
>
> Our focus on second-order elliptic PDEs in the manuscript is primarily for clarity and concreteness. Without specifying a concrete PDE form, it would be difficult to precisely express the structure of the covariance matrix (associated with the diffusion term in the SDE), which would make the continuous framework overly abstract and less accessible to readers.
>
> Assumption 1 in our paper is introduced mainly to ensure that the coefficients of the PDE are bounded, so that certain terms in the loss function—those dependent only on the PDE coefficients and independent from the network parameters—can be properly controlled. For other differential operators, if a reader is interested in extending our results to their SDE formulation in the PINN framework, they can easily check whether their coefficients satisfy the boundedness condition, and then follow the same theoretical derivations as we provided.
>
> As stated in the paper, our use of second-order elliptic equations is not intended to restrict the applicability of our results, but rather to make the presentation of the SDE structure and our results clearer. Theorem 1 holds as long as the coefficients of the differential operator are bounded, which applies to a wide range of PDEs beyond the examples explicitly shown.
>
> We appreciate your suggestion and will clarify this point explicitly in the abstract and throughout the main text in our revised version. Thank you again for your constructive feedback.
>
> > ### Weakness 2:
> >
> > "... a continuous framework" is also too generic for the title (use "continuous-time" instead?); it is also challenging to read about "continuous time" frameworks while the PDEs in question are not time-dependent, yet the introduction and abstract make no distinction about this.
>
> ### Reply to Weakness 2:
>
> Thank you for your suggestion regarding the title and for highlighting the potential confusion around the use of the term "continuous" in our context.
>
> To clarify, when we refer to a "continuous framework," we mean the continuous-time modeling of the training dynamics for neural network-based PDE solvers—not the time dependence of the PDEs themselves. Specifically, our work approximates the discrete sequence of parameters generated by SGD with a continuous-time stochastic process as the step size tends to zero, analogous to the relationship between gradient flow (a continuous-time dynamical system) and discrete gradient descent. This framework is independent of whether the underlying PDE is time-dependent or stationary.
>
> Regarding the generality of our title: previous works on continuous-time modeling of SGD required the loss gradients to be globally Lipschitz—a strong condition that is generally not satisfied for most neural network-based PDE solvers, except in rare cases such as linear networks. Our approach removes this limitation and can therefore be applied to a much broader range of optimization algorithms and network architectures. We chose a broader title to emphasize this theoretical advance, but we appreciate your feedback and will revise the abstract to more clearly reflect our contributions and avoid any ambiguity.
>
> Thank you again for your helpful comments.

---

> ### Author Response · Authors · 2025-11-20
>
> > ### Weakness 3:
> >
> > The authors claim that "..our results readily extend to more general equations and deeper network architectures"....
> ### Reply to Weakness 3:
>
> Thank you for raising this point and allowing us to clarify our statements regarding the generality of our framework.
>
> The main technical challenge addressed in our work is establishing the well-posedness (existence and uniqueness) of the SDE that approximates the SGD dynamics when the loss gradient is not globally Lipschitz—an assumption that rarely holds for nonlinear or deep neural networks. To overcome this, we construct a Lyapunov function (as shown in Lemma 3 in the appendix), and by leveraging SDE theory (see Theorem 2), we guarantee the well-posedness of the limiting SDE under these general conditions.
>
> The reason our results can be extended to more general equations, deeper networks, and empirical losses is that, regardless of these choices, the Lyapunov function we use (the regularized loss plus any fixed positive constant, as in Lemma 3) remains valid for controlling the dynamics of the resulting SDE. As long as this function can be constructed for the modified setting, the same proof techniques apply, and weak error estimates can be established using standard arguments.
>
> We agree that providing explicit theoretical demonstrations for all such extensions would further strengthen our claims. Thank you again for your constructive feedback.
>
> > ### Weakness 4:
> >
> > Similarly, in the title and the main text, "dnn-based PDE solvers" are mentioned, but in the setting (e.g. eq. 2), only shallow, two-layer networks are considered....
> ### Reply to Weakness 4:
>
> Thank you for highlighting this important distinction between shallow and deep neural networks.
>
> We agree that shallow and deep networks may exhibit different behaviors and properties in practice. In our work, the focus on two-layer networks was primarily motivated by the need to present the SDE formulation and theoretical results in a clear and concrete manner, without making the presentation overly abstract or technical. However, as discussed in our response to your previous points, the key arguments—especially those related to the construction of a Lyapunov function for the limiting SDE—extend to deep architectures, provided similar regularity and boundedness conditions are satisfied.
>
> Thank you again for your thoughtful comments.
>
> > ### Weakness 5:
> >
> > The computational experiments do not show the performance of SGD on a PDE, but just an ODE....
>
> ### Reply to Weakness 5:
>
> Thank you for your comment. In response, we have added an experiment on the Helmholtz and Allen--Cahn equations, with results consistent with those in Section 4. Specifically, we observe that near minimizers with low sharpness, GD exhibits greater stability to the choice of step size compared to SGD, whereas the opposite occurs near minimizers with very high sharpness. The detailed setup and results have been included in the appendix.
>
> The original focus on cases with closed-form solutions is mainly for convenience, as it allows us to explicitly select and analyze desired minimizers for the experiments. Previous studies [1] have shown that the stability region of SGD with respect to step size is generally smaller than that of GD, which matches our observations near less sharp minimizers. This behavior is expected to extend to general PDE settings.
>
> However, the step size stability of SGD observed near highly sharp minimizers does not align with prior conclusions, and we believe this unexpected phenomenon is interesting and worth further investigation in future work.

---

> ### Author Response · Authors · 2025-11-20
>
> > ### Weakness 6:
> >
> > For many PINNs the optimizer starts with Adam iterations, followed by additional iterations with L-BFGS....
>
> ### Reply to Weakness 6:
>
> Thank you for pointing this out. We agree that in practical PINN training, it is indeed common to first use Adam and then switch to L-BFGS for further optimization.
>
> For Adam, while some works have studied its continuous-time approximation in supervised learning, these analyses rely on the restrictive assumption of globally Lipschitz loss gradients. Our framework provides a potential way to relax this requirement and extend continuous-time modeling to Adam under weaker conditions, which we plan to investigate in future work.
>
> As for L-BFGS, to our knowledge, there has been little work on its continuous-time approximation, and developing such a theory could be an interesting research direction moving forward.
>
> In the revised manuscript, we have added a limitations section where we explicitly acknowledge and discuss these points.
>
> Finally, regarding our use of “adaptively switching optimizers” in the abstract: this is not meant to claim the use of adaptive optimizers like Adam, but rather to suggest that using different optimizers in different training phases—such as near sharp or flat minima—can yield better results. This observation is based on our experiments in Section 4, where we analyze the sensitivity of SGD and GD to step size around various critical points.
>
> Thank you again for your helpful suggestions.
>
> > ### Weakness 7:
> >
> > The manuscript lacks a reflection on the limitations of the current work.
>
> ### Reply to Weakness 7:
>
> Thank you for your suggestion. In the revised version, we have added a dedicated section discussing the main limitations of our work to provide a more balanced and transparent perspective. We appreciate your feedback and believe it helps improve the clarity and completeness of our manuscript.
>
> > ### Weakness 8:
> >
> >Minor remarks....
>
> ### Reply to Weakness 8:
> Thank you for your minor remarks. We have addressed them and made the corresponding changes in the revised manuscript.
>
> ## Replies to Questions
>
> > ### Question 1:
> >
> > Other work (Li et al. 2017, 2019) considers not only SGD but also SGD with momentum and Nesterov's method. Is an extension of your work possible for these variants of SGD?
>
> ### Reply to Question 1:
>
> Thank you for your question and for bringing up these important variants.
>
> Our framework can indeed be extended to SGD with momentum and Nesterov’s method. As we emphasized before, the key innovation of our work is establishing the well-posedness of the noisy regularized SDE as a continuous-time approximation, even when the loss function does not have globally Lipschitz gradients—a common situation in neural PDE solvers. By constructing suitable Lyapunov functions, our approach goes beyond the limitations of previous works, which required strong global Lipschitz assumptions.
>
> This methodology is flexible and can be adapted to other optimization algorithms, including those you mentioned, as well as to a variety of PDEs and solver architectures. We will clarify this point in the revised manuscript. Thank you again for your insightful suggestion.
>
> >### Question 2:
> >
> > The loss function for PINNs often includes additional terms (e.g. boundary/initial conditions, additional physical constraints, enforcing symmetry)....
>
> ### Reply to Question 2:
>
> Thank you for your insightful question.
>
> As discussed in Remark 1 of our manuscript, if the additional loss terms (such as those for linear boundary or initial conditions) are quadratic, they can be treated similarly to standard supervised learning losses, and previous theoretical results apply directly.
>
> For more general constraints—such as non-quadratic losses from physical constraints or symmetry enforcement—the essential consideration is whether these new terms allow the construction of a suitable Lyapunov function. If so, the main results of our paper, including Theorem 1, continue to hold without significant modification. In practice, this often requires the added terms to ensure sufficient regularity and growth control of the total loss.
>
> Thank you once again for your careful review and valuable feedback on our manuscript. Your insightful questions and suggestions have greatly helped us improve the clarity and rigor of our work. If you have any further questions or concerns, we warmly welcome your continued input and look forward to further discussion. Thank you for your time and support.
>
> [1] Lei Wu, Chao Ma, and Weinan E. How SGD selects the global minima in over-parameterized learning: A dynamical stability perspective.Advances in Neural Information Processing Systems, 31, 2018.

---

### Meta-Review · Area_Chair_xzt5 · 2025-12-29

**Summary:**

The paper received three reviews with scores ranging from rejection to marginally below acceptance. The major concerns shared among the reviews include the following:
1) Multiple reviewers believed that the assumptions and restrictions introduced in the theoretical analysis are too strong, limiting the practical applicability of the results.
2) Many reviewers pointed out that the experiments reported empirical validation on simple test problems only, particularly one-dimensional ODEs with small neural network models. Its extensions to PDEs and modern network models were not obvious to some reviewers.
3) Several reviewers suggested a comprehensive comparison with and discussion on more optimizers besides SGD, e.g., variants of Adam and L-BFGS.

Reviewers also raised individual concerns about the paper’s writing, technical details, and position in the literature.

**Reviewer Concerns:**

For the first concern, the rebuttal clarified the context of these theoretical assumptions in practical applications, emphasizing their significance for theoretically rigorous results. I think the rebuttal partially addressed this concern by providing empirical evidence that some of these assumptions can hold in practical settings.

For the second concern, the rebuttal stated that the results can be extended to PDEs and deeper networks, provided the assumptions hold. The rebuttal provided additional experiments on two PDEs (Helmholtz and Allen-Cahn), which I think partially addressed this concern. Having more validations on deeper networks would help further clarify the concerns with small networks.

Regarding the last concern, the rebuttal reiterated its theoretical perspective and argued that such comparisons are unnecessary for the main objective of this work. After reading the reviews and the rebuttal, it remains unclear whether this argument can fully address the concern.

**Reviewer Scores:**

I think the rebuttal would positively affect some reviewers’ opinions of the paper, given the clarifications summarized in the questions above. However, some major concerns would require more substantial updates, e.g., additional theoretical and empirical work to justify the assumptions in practice and more substantial experiments on PDEs and other baselines. I think such updates would require more effort than what a typical reviewer-author discussion could cover. Therefore, based on the current form of the reviews and rebuttal, I am not optimistic enough to recommend acceptance of the paper.

---

### Decision · Program_Chairs · 2026-01-26

Reject